# A tectonic-rules based mantle reference frame since 1 billion years ago – implications for supercontinent cycles and plate-mantle system evolution

R. Dietmar Müller[1], Nicolas Flament[2] John Cannon[1], Michael G. Tetley[3], Simon E. Williams[4], Xianzhi Cao[5], , Ömer F. Bodur[2], Sabin Zahirovic[1], and Andrew Merdith[6]

[1]EarthByte Group, School of Geosciences, The University of Sydney, NSW 2006, Australia
[2]GeoQuEST Research Centre, School of Earth and Environmental Sciences, University of Wollongong, Northfields Avenue, NSW 2522, Australia
[3]University of Texas Institute for Geophysics, Jackson School of Geosciences, The University of Texas at Austin, Texas, 78758, United States
[4]Department of Geology, Northwest University, Xi'an, 710069, China and EarthByte Group, School of Geosciences, The University of Sydney, NSW 2006, Australia
[5]Frontiers Science Center for Deep Ocean Multispheres and Earth System; Key Lab of Submarine Geosciences and Prospecting Techniques, MOE and College of Marine Geosciences, Ocean University of China, Qingdao 266100, China
[6]School of Earth and Environment, University of Leeds, Leeds, UK.

*Correspondence to*: R. Dietmar Müller (dietmar.muller@sydney.edu.au)

**Abstract.** Understanding the long-term evolution of Earth's plate-mantle system is reliant on absolute plate motion models in a mantle reference frame, but such models are both difficult to construct and controversial. We present a tectonic rules-based optimisation approach to construct a plate motion model in a mantle reference frame covering the last billion years and use it as a constraint for mantle flow models. Our plate motion model results in net lithospheric rotation consistently below 0.25°/Myr, in agreement with mantle flow models, while trench motions are confined to a relatively narrow range of -2/+2 cm/yr since 320 Ma, during Pangea stability and dispersal. In contrast, the period from 600 Ma to 320 Ma, nicknamed here the "zippy tricentenary", displays twice the trench motion scatter compared to more recent times, reflecting a predominance of short and highly mobile subduction zones. Our model supports an orthoversion evolution from Rodinia to Pangea with Pangea offset approximately 90° eastwards relative to Rodinia—this is the opposite sense of motion compared to a previous orthoversion hypothesis based on paleomagnetic data. In our coupled plate-mantle model a broad network of basal mantle ridges forms between 1000 and 600 Ma, reflecting widely distributed subduction zones. Between 600 and 500 Ma a short-lived degree-2 basal mantle structure forms in response to a band of subduction zones confined to low latitudes, generating extensive antipodal lower mantle upwellings centred at the poles. Subsequently the northern basal structure migrates southward and evolves into a Pacific-centred upwelling while the southern structure is dissected by subducting slabs, disintegrating into a network of ridges between 500 and 400 Ma. From 400 to 200 Ma, a stable Pacific-centred degree-1 convective planform emerges,. It lacks an antipodal counterpart due to the closure of the Iapetus and Rheic oceans between Laurussia and Gondwana as well as due to coeval subduction

between Baltica and Laurentia and around Siberia, populating the mantle with slabs until 320 Ma when Pangea is assembled. A basal degree-2 structure forms subsequent to Pangea breakup, after the influence of previously subducted slabs in the African hemisphere on the lowermost mantle structure has faded away. This succession of mantle states is

distinct from previously proposed mantle convection models. We show that the history of plume-related volcanism is consistent with deep plumes associated with evolving basal mantle structures. This Solid Earth Evolution Model for the last 1000 million years (SEEM1000) forms the foundation for a multitude of spatio-temporal data analysis approaches.

**Short summary.** We introduce a community model for the evolution of the Earth's plate-mantle system. Created with
open-source software and an open-access plate model, the model covers the last billion years, including the formation, breakup and dispersal of two supercontinents, and the creation and destruction of numerous ocean basins. The model allows us to 'see' into the Earth in 4D, and helps unravel the connections between surface tectonics and the "beating heart" of the Earth, its convecting mantle.

## 1 Introduction

### 1.1 Relative versus absolute plate motions

Plate tectonics unifies our understanding of the evolving solid Earth, the ocean basins, landscapes, and the evolution of life. Since the advent of plate tectonic theory enormous progress has been made in mapping the relative motions of the plates through time, constrained by magnetic anomaly and fracture zone data in the ocean basins, and a variety of geological, geophysical and paleomagnetic data on the continents (see summary by Cox and Hart, 2009).

Nonetheless, absolute plate motions, i.e. the motions of the plates relative to a fixed reference system, such as the spin axis of the Earth or the mesosphere, have been much more difficult to constrain. Both the paleolatitude of a plate as well as its paleo-meridian orientation can be calculated using paleomagnetic data, providing a paleomagnetic pole for a given plate (Cox and Hart, 2009). However, since the Earth's magnetic dipole field is radially symmetric, paleo-longitudinal information cannot be determined from paleomagnetic data alone, unless further assumptions are made (Torsvik and

Cocks, 2019). For relatively recent geological times (Late Cretaceous to present), seamount chains as well as continental volcanic formations with a linear age progression can be used to restore plates to their paleo-positions (including paleo-latitude and paleo-longitude), with the assumption that surface hot spots resulting from intersections of mantle plumes with the surface, are either fixed relative to each other or moving slowly with respect to each other (Koppers et al., 2021). Various alternative time-dependent regional and global absolute plate motion models based on hotspot tracks have been

developed over the past decades, with some based on hotpot track data alone (e.g., Maher et al., 2015; Wessel and Kroenke, 2008) while others reflect a combination of relative plate motion and constraints provided by mantle convection models (e.g., O'Neill et al., 2005; Steinberger, 2000). Hotspot track-based models for recent geological times can be combined with models based on paleomagnetic data for earlier times, forming "hybrid models" (Torsvik et al., 2008).

The difficulties involved in constructing hotspot reference frames, and their lack of robustness for pre-Cretaceous times,
reflecting a shortage of preserved age-dated hotspot tracks, led to the idea of a subduction reference frame. This follows
the assumption that slabs sink vertically through the entire mantle, allowing the location of past subduction zones to be
reconstructed based on global mantle tomographic models (Van Der Meer et al., 2010). However, the empirical
"longitudinal correction" applied to the plates in such models differs significantly with plate positions derived from
hotspot track data (Butterworth et al., 2014). Domeier et al. (2016) tested the concept of a subduction reference frame
concept using a range of tomographic models and concluded that the method may be used for reconstructions back to
130 Ma, reflecting imaged slabs down to a depth of 2300 km.

**1.2 Large Low Shear Velocity Provinces as longitudinal markers?**

Considering that neither age-progressive hotspot tracks nor subducted slabs are useful for reconstructing the
past positions of plates before the Cretaceous Period, and the considerable challenge of reconstructing paleolongitude
from paleomagnetic data, Torsvik and Cocks (2019) built on the idea of Large Low Shear Velocity Province (LLSVP)
stability put forward in Burke and Torsvik (2004). LLSVPs were regarded as useful in this context as their edges were
proposed to act as "plume generation zones", offering an avenue to align age-dated large igneous provinces (LIPs) and
kimberlites with the present-day edges of LLSVPs. This hypothesis is built on the assumption that LIPs and kimberlites
are the product of plumes rising from LLSVP boundaries, which remain stationary through time (Burke and Torsvik,
85    2004).
However, using statistical approaches a number of studies (e.g., Austermann et al., 2014; Davies et al., 2015a) have
shown that this correlation is not robust, whilst a follow up study by Doubrovine et al. (2016) essentially confirms this in
the sense that one cannot conclusively state that plumes are forming at LLSVP edges versus interiors. Using similar
statistical approaches, Flament et al. (2022) recently showed that that the alignment of LIPs and kimberlites is statistically
as consistent with the boundaries and interiors of mobile basal mantle structures shaped by Earth's reconstructed
subduction history as with fixed ones.
This plume generation zone method offers a reproducible and quantifiable method of adding a longitudinal correction to
reconstructed plates, thus providing an apparent solution to reconstructing longitude. Le Pichon et al. (2019) also built
an absolute reference frame based on an assumption of stationary deep mantle structures, back to 400 Ma. However, the
basic tenet of these approaches, namely the long-term stability of LLSVPs, has been challenged. Recent mantle
tomographic images, combined with fluid mechanic constraints, have resulted in a view that LLSVPs are composed of
bundles of thermochemical upwellings enriched in denser than average material (Davaille and Romanowicz, 2020), an
interpretation that follows numerous previous papers coming to similar conclusions (e.g., Garnero and Mcnamara, 2008;
Heyn et al., 2018; Tan et al., 2011). Only when tomographic models are filtered to long wavelengths do these structures
take on the appearance of homogenous, uniform and potentially stable provinces (see also Schuberth et al., 2009; Tkalčić
et al., 2015). These models and observations, as well as mantle flow models (e.g., Zhang et al., 2010; Zhong and Liu,

2016; Cao et al., 2021a; Davies et al., 2015a; Garnero and Mcnamara, 2008; Flament et al., 2017; Bull et al., 2014; Davies et al., 2012), indicate that the shape of LLSVPs is controlled by the distribution of subducted slabs and the position of LLSVPs relative to them, implying that LLSVP structures and their boundaries are mobile. Based on mantle flow models,

Zhang et al. (2010) concluded that the African LLSVP is unlikely to have existed in its current form before 230 Ma, while Mitchell et al. (2012) suggested, based on distribution patterns of virtual geomagnetic poles, that neither the African nor the Pacific antipodal upwellings existed before the creation of Pangea. Additionally, Doucet et al. (2020b) used the geochemical composition of plume-related basalts to argue for a dynamic relationship between deep mantle structures and plate tectonic evolution. These inferences remain to be further tested, but the apparent unlikelihood of LLSVP

stability over long geological time periods challenges the usefulness of the method proposed by Torsvik and Cocks (2019) as a universal solution for reconstructing the longitude of plates.

### 1.3 Alternative modes of supercontinent formation

Possible alternative modes of supercontinent formation include: (1) closing of the youngest ocean basin on the same hemisphere as the last supercontinent ("introversion", re-closing the Atlantic Ocean from the present configuration),

(2) closing of the older antipodal ocean basin ("extroversion", closing the Pacific Ocean from the present configuration) and (3) closing an ocean basin orthogonal to the direction of opening of the last ocean basin ("orthoversion", e.g. closing the Arctic Ocean from the present configuration) (Evans et al., 2016; Murphy and Nance, 2003; Murphy et al., 2009). Following these ideas, Mitchell et al. (2012) proposed an alternative method to obtain paleo-longitude from paleomagnetic data across supercontinent cycles. They utilised the record of oscillatory true polar wander (TPW) as

expressed in apparent polar wander paths. True Polar wander that is a solid-body rotation of the Earth about the equatorial minimum moment of inertia with respect to its spin axis causing geographic poles to "wander" (Raub et al., 2007). Mitchell et al. (2012) suggested that consecutive supercontinents are roughly separated from each other by 90° of longitude. This orthoversion reconstruction effectively assembles a new supercontinent above one of the downwelling subduction girdles surrounding the previous supercontinent. While this method provides a conceptual model for absolute

plate motions, it falls short of being useful for deriving an actual mantle reference frame through time (Torsvik and Cocks, 2019). Specifically, the practical application of this method is limited due to uncertainties related to the prediction of the occurrence of true polar wander episodes rather than deriving them independently and the inability of the method to uniquely determine whether the path from one supercontinent to the next is from west to east or vice versa. To understand the geodynamic history of continents after supercontinent breakup it is imperative to have both a continuous time series

of absolute plate motions and to have a constraint on whether the western or the eastern borders of a dispersing supercontinent move across a major downwelling driven by slabs sinking in the mantle.

**1.4 Net lithospheric rotation and trench migration**

The net rotation of the lithospheric shell of the Earth relative to the underlying mantle owes its origin to lateral variations in upper mantle viscosity and mantle structure (Rudolph and Zhong, 2014; Ricard et al., 1991). Many published absolute plate motion models suffer from plate velocity artefacts, typically resulting in excessive net lithospheric rotation magnitudes. Often, absolute plate motion models are based on fitting geological observations which in some instances result in either the over-fitting of observations, or fitting the wrong trends within data from volcanic chains (see Schellart et al., 2008, for a discussion), resulting in resulting in geodynamically problematic models that are difficult to reconcile with our knowledge of mantle rheology (e.g., Rudolph and Zhong, 2014). As a consequence, mantle flow modellers often convert a plate tectonic model into a so-called no-net-rotation (NNR) reference frame (e.g. Mao and Zhong, 2021), in which the net rotation of the entire lithosphere relative to the mantle is set to zero at all times. Upper magnitude limits to net lithospheric rotation have been proposed based on mantle flow modelling, suggesting net rotation should be a positive, non-zero value less than ~0.2-0.3°/Myr (Becker, 2006; Conrad and Behn, 2010), but not necessarily zero. Using a low net rotation threshold for building an absolute plate motion model, as opposed to assuming that it is zero, implicitly acknowledges the heterogeneous nature of the lithosphere and variations in upper mantle viscosity, an often-cited criticism of NNR reference frames (Le Pichon et al., 2019). Here we consider a NNR reference frame as an end member, while focussing on building an absolute plate motion model that limits net rotation to comply with geodynamic constraints (Becker, 2006; Conrad and Behn, 2010). Williams et al. (2015) analysed a set of alternative absolute plate motion models and proposed that global optimization of trench migration characteristics should be considered as an additional criterion in the construction of absolute plate motion models, a strategy that we include here. Williams et al. (2015) followed the insights of Schellart et al. (2008) who observed that most trenches mostly roll back slowly at speeds of ~0-2 cm/yr at present-day, with trench advance being extremely rare.

**1.5 Alternative approach for mantle reference frame construction**

All absolute reference frames discussed above fall in the category of mantle reference frames, i.e. they are designed to estimate the position of plates relative to the mantle through time, as opposed to the spin axis. Unlike the spin axis, the convecting mantle, does not provide a stable, fixed reference system through time. A mantle reference frame attempts to isolate the motions of plates relative to the mantle, given as plate rotations relative to the Earth's spin axis, which is assumed to be fixed. Such a reference frame is therefore agnostic of TPW. Paleomagnetic data record information informing both the motions of the plates relative to the mantle as well as TPW, and can thus be used to restore the plates in terms of their "true" latitudinal positions through time, which is useful for paleoclimate studies. However, when reconstructed paleomagnetic poles derived from paleomagnetic data cannot constrain paleolongitude due to the radial symmetry of the Earth's magnetic field (Cox and Hart, 2009), and therefore cannot be used to accurately track the east-west movement of the plates across mantle upwellings and downwellings, unless additional assumptions

are made—see Section 1.2 in Torsvik and Cocks (2019). In contrast, an ideal mantle reference frame provides both constraints on both paleolatitudes and paleolongitudes of plates relative to the mantle. However, as it does not consider TPW, it does not provide paleogeographic reconstructions useful for paleoclimate studies (Van Hinsbergen et al., 2015). These two types of reference frames are complementary to each other.

To overcome the limitations of traditional mantle reference frames, Tetley et al. (2019) presented a new method applying a joint global inversion to evaluate the contribution of multiple time-dependent absolute plate motion constraints including fit to age-progressive hotspot tracks, optimizing subduction zone migration behaviours and minimizing rates of net lithospheric rotation. This approach explicitly excludes true polar wander, as the method is deliberately aimed at reconstructing the plates relative to the convecting mantle. The method automatically provides both paleo-latitudes and paleo-longitudes relative to the mantle, thus providing a mantle reference frame, expressed as rotations of the plates relative to the spin axis of the Earth which is assumed to be stationary. This approach has been extended for the application in this paper by including evaluation of continental velocities relative to the mantle as additional criterion. Tectonic rules-based plate motion model optimisation can be applied to any plate motion model with continuous closing plate boundaries through time (Gurnis et al., 2012).

Our aim is to derive a mantle reference frame for the plate motion model of Merdith et al. (2021), extending the tectonic-rules based approach proposed by Tetley et al. (2019) to the last billion years. This results in a "deep-time" plate motion model suitable for plate-mantle system simulations, and allows us to test the orthoversion hypothesis suggested by Mitchell et al. (2012) independently of any reliance on paleomagnetic data. It also allows us to evaluate the difference between the widely used NNR reference frame approach and a more complex application of tectonic rules to reference frame construction, aiming to minimise net rotation jointly with other key parameters. Lastly, it allows us to design a plate-mantle system model to understand how the deep mantle structure responds to plate motions following a set of tectonic rules. For instance, we can test the hypothesis by Mitchell et al. (2012) that the African and Pacific LLSVPs did not exist before Pangea assembled.

## 2 Methods

### 2.1 Mantle reference frame optimisation

It needs to be stated in the outset that prior to the assembly of Pangea we have much less constraints on the relative positions of plates as compared to more recent times. To render mantle reference frame construction tractable, we leave relative plate motions unaltered and focus on optimising a single, global reference frame. Our workflow for absolute plate motion model construction follows the iterative method outlined in Tetley et al. (2019) (Fig. 1). For a given iteration, the approach starts with perturbing an initial absolute Euler rotation (pole latitude, pole longitude and angle

magnitude) for a given reference continent or plate, and then calculates a series of fit metrics with selected constraining data using objective (or cost) functions. This process continues until a global minimum is found. For this study, we use continental Africa as the reference (as it forms the base of the plate model rotation tree of Merdith et al. (2021)). Following Tetley et al. (2019), we calculate fit metrics computed from evaluating (1) net lithospheric rotation rate (NR), (2) trench migration rate (TM), and (3) the fit of present-day hotspots to the major age-progressive hotspot tracks for the period of 0-80 Ma only (HS). In addition to the above, we extend the existing method to also compute a fourth constraining criterion: (4) median global continental absolute plate velocity (PV). We introduce continental absolute plate velocities as additional criterion to prevent mean oceanic plate velocities based on synthetic plates from potentially inducing unreasonably high continental speeds globally, as the deep-time reconstructions used here include large swathes of reconstructed ocean floor that is now subducted, based on a variety of indirect pieces of geological evidence (Merdith et al., 2021).

The four constraining criteria are applied to the absolute plate motion model optimization with the following assumptions/bounds: (1) rates of net lithospheric rotation (NR) are minimized but non-zero, (2) global trench migration velocities are minimized, favouring trench retreat over trench advance, (3) spatio-temporal misfit between plate motion model and present-day hotspot chains is minimized, and (4) global continental median plate speed remains < 60 mm/yr , based on continental plate speed statistics reported in Zahirovic et al. (2015). The contribution of individual optimization parameters to the overall inversion are initially scaled by relative magnitude and then weighted by empirically determined weights. For times older than 80 Ma, NR=1, TM=0.5, PV=0.5, HS=0 (0-80 Ma, NR=TM=PV=HS=1). From this optimised plate motion model, we then reconstruct the age-area distribution of the ocean floor, based on the evolving plate boundary topologies and rotations, following the method by Williams et al. (2021).

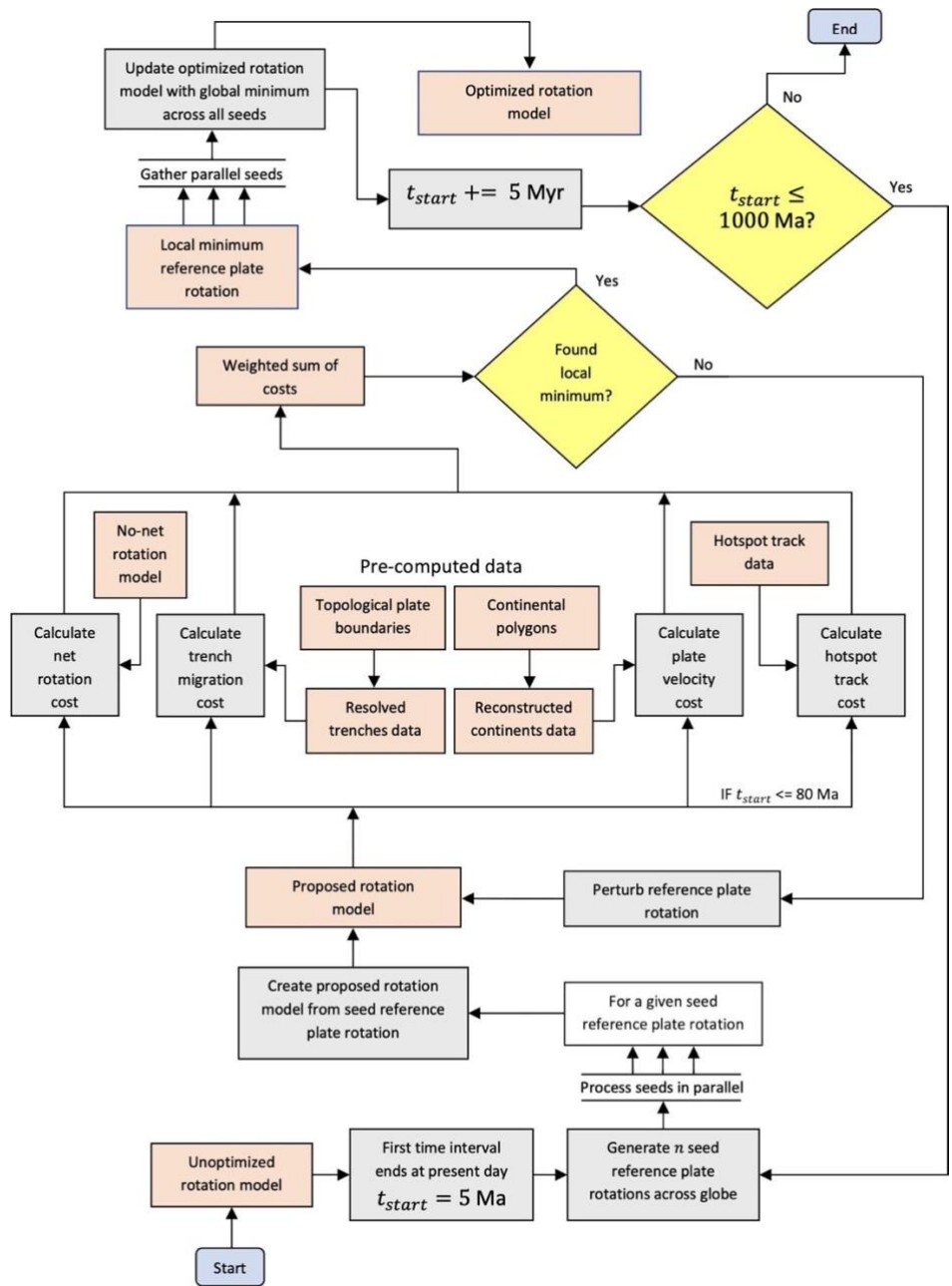

Figure 1. Optimization workflow including decisions (yellow diamonds) and directing flow through processes (rectangular grey boxes) that accept input and produce output data (orange boxes). The beginning and end of the workflow are denoted by light blue boxes with rounded edges. The workflow sequentially optimizes absolute plate motion in 5 Myr time intervals starting at present day and progressing backwards in time until 1000 Ma. Within each time interval the motion of a reference plate (and thereby the absolute motion of all plates) is optimized by perturbing its rotation while

iteratively minimizing the cost of an objective function. The reference plate is Africa between 550-0 Ma and Laurentia between 1000-550 Ma. Global optimisation in the current time interval is initiated by generating 400 rotations with which to seed local optimizations. Each seed is generated from the reference plate rotation optimized in the preceding time interval by retaining its rotation angle but distributing its rotation pole (latitude and longitude) to 400 uniform locations across the globe. To take advantage of parallel processing we distribute these 400 seeds in parallel across multiple computational nodes, with each node performing a local optimization of a single seed, with the results from all nodes gathered to find the globally minimal reference plate rotation for the current time interval. The objective function (minimized during optimisation) consists of four separate weighted cost functions using the perturbed rotation model as input: (1) misfit distances of hotspot trails (between 0-80 Ma), (2) net rotation of reference plate with an extra penalty if below 0.08 or above 0.20 degrees/Myr (for efficiency the net rotation is calculated relative to a no-net rotation model), (3) trench migration calculated as mean magnitude of trench-orthogonal velocity sampled uniformly along all trenches with an extra penalty if the mean magnitude exceeds 30 mm/yr (for efficiency we only resolve trenches from topological plate boundaries once per time interval), and (4) plate velocity magnitude calculated as median velocity of uniformly sampled points inside continents with an extra penalty if magnitude exceeds 60 mm/yr (for efficiency we only reconstruct continental polygons and their contained sample points once per time interval).

## 2.2 Mantle convection modelling

We model mantle convection using the extended-Boussinesq approximation in a version of CitcomS (Zhong et al., 2008) which has been modified for progressive assimilation of surface boundary conditions from plate reconstructions (Bower et al., 2015). We build the thermal structure of the lithosphere using reconstructed seafloor ages and a half-space cooling model with maximum seafloor age set to 80 Myr. This corresponds to a fast and simple implementation of the equivalent of a plate model; for the purpose of our application, the difference to using an actual plate model would be negligible. Similarly, we build the thermal structure of subducting slabs from the surface to 350 km depth with a dip angle of 45° using seafloor ages at 1 Myr intervals. We apply an isothermal (T = 273 K) and kinematic (plate velocities exported from the reconstructions) boundary condition at the surface, and an isothermal (T = 3373 K) and free-slip boundary condition at the CMB. Slabs are initially built from surface to 1000 km depth, with dip angles of 45° above 425 km depth and 90° below.

We consider 4 mantle model cases: cases OPT1 and OPT2 use our optimized reconstruction as time-dependent boundary conditions, case PMAG uses the reconstruction from Merdith et al. (2021) which is in a paleomagnetic reference frame, and case NNR uses the same reconstruction except with net lithospheric rotations removed (i.e., a no-net-rotation reference frame). The initial condition includes a 113-km-thick denser basal layer. The excess density is defined by the buoyancy ratio $B = \delta\rho_{ch}/(\rho\alpha\Delta T)$, where $\rho$ is the density, $\alpha$ is the coefficient of thermal expansivity, $\Delta T = 3100$ K is the temperature difference across the mantle, and $\delta\rho_{ch}$ is density contrast disregarding thermal effects. The buoyancy ratio B is 0.25 for case OPT1, and 0.325 for other cases (Table 1), which respectively corresponds to an excess

density of about ~1% ($\delta\rho_{ch} = 56.8$ kg m$^{-3}$) and ~1.3% ($\delta\rho_{ch} = 73.8$ kg m$^{-3}$) for the basal layer, if we take $\rho = 5546$ kg m$^{-3}$ (the average value of the bottom 100 km above the CMB from Preliminary Reference Earth Model (Dziewonski and

260 Anderson, 1981)), and $\alpha = 1.32 \times \mathbf{10^{-5}}$ K-1 (the average value of the bottom 100 km above the CMB. The composition field is tracked with tracers using the ratio tracer method (Mcnamara and Zhong, 2004; Tackley and King, 2003). Before the main calculation, the 1000 Ma plate configuration was applied during a 250 Myr warm-up phase.

The convective vigour is controlled by the Rayleigh number: $Ra = \alpha_0\rho_0 g_0 \Delta T h_M^3 / \kappa_0\eta_0 = 7.8 \times 10^7$ , where

$\alpha_0 = 3 \times 10^{-5}$ K$^{-1}$ is the reference coefficient of thermal expansivity at the surface, $\rho_0 = 4{,}000$ kg m$^{-3}$ is the density, $g_0 = 9.81$ m s$^{-2}$ is the acceleration of gravity at the surface, $h_M = 2{,}867$ km is the thickness of the mantle and $\kappa_0 = 1 \times 10^{-6}$ m$^2$ s$^{-1}$ is the thermal diffusivity. The dissipation number is $Di = \alpha_0 g_0 R_0 / C_{P_0} = 1.56$, where $C_{P_0} = 1{,}200$ J kg$^{-1}$ K$^{-1}$ is the reference heat capacity. The rate of internal heating for the whole model is $H = 33.6$ TW. Viscosity is temperature, composition and depth dependent:

$$\eta = \eta(r)\,\eta_0\,\eta_C\exp\left(\frac{E_\eta + \rho_0 g Z_\eta(R_0 - r)}{R(T + T_{off})} - \frac{E_\eta + \rho_0 g Z_\eta(R_0 - R_c)}{R(T_{CMB} + T_{off})}\right)$$

where $\eta(r)$ is a depth-dependent pre-factor with values 0.02, 0.002, 0.02, 0.2 for mantle above 160 km, between 160-310 km, between 310-660 km and below 660 km, respectively. $\eta_0 = 1.1e21$ Pa s is the reference viscosity, $\eta_C$ is the

275 compositional viscosity pre-factor: 1, 100, 10 for ambient mantle, continental lithosphere and basal layer, respectively, in the initial condition. $E_\eta = 283.5$ kJ mol$^{-1}$ is the activation energy, $Z_\eta = 2.1$ cm$^3$ mol$^{-1}$ is the activation volume, $g$ is the acceleration of gravity, $R_0 = 6{,}371$ km is the radius of the Earth, $r$ is the radius, $R = 8.31$ J mol$^{-1}$ K$^{-1}$ is the universal gas constant, $T$ is the temperature, $T_{off} = 496$ K is a temperature offset, and $R_C = 3{,}504$ km is the radius of the core. $E_\eta$ and $T_{off}$ are selected to obtain viscosity variations by three orders of magnitude as a function of temperature (Flament, 2019). The

280 model consists of ~13 million nodes (129 $\times$ 129 $\times$ 65 $\times$ 12), with radial mesh refinement to obtain slightly higher resolutions at the surface (~50 $\times$ 50 $\times$ 15 km) and CMB (~28 $\times$ 28 $\times$ 27 km), and a lower resolution in the mid-mantle (~40 $\times$ 40 $\times$ 100 km).

## 2.3 Assessing the match of volcanic eruption locations and basal mantle structures

We follow the approach of Flament et al. (2022) to evaluate model success from (i) the time-dependent match

between volcanic eruption locations and basal mantle structures, and (ii) the match between the present-day mantle structure predicted by mantle flow models and imaged by tomographic models. We are primarily interested in basal mantle structures (BMSs) that are hot in mantle flow models and slow in tomographic models. The first step consists of carrying out a cluster analysis of lower mantle structure. As in Flament et al. (2017), we classify ~ 200,000 equally-

spaced points on Earth's surface into two groups of points with similar variations in a given property between 1000-2800
km depth (Flament et al., 2017; Lekic et al., 2012) . The analysis was carried out on both mantle flow and tomographic
models using the scientific Python implementation of the *k*-means (Macqueen, 1967) algorithm
(http://docs.scipy.org/doc/scipy/reference/generated/scipy.cluster.vq.kmeans2.html), deriving four metrics from these
clusters:

(i)    The fractional area $f_a$ of deep mantle covered by BMSs hot basal mantle structures in cluster space. $f_a$ is time-averaged for the flow models. The purpose of this metric is to verify that the area of predicted BMSs is consistent with the area of imaged BMSs.

(ii)    The accuracy $Acc = (TP+TN)/A$ (Flament, 2019; Flament et al., 2017) that quantifies the match between present-day BMSs from flow models and BMSs from seven S-wave tomographic models: SAW24B16 (Mégnin and Romanowicz, 2000); HMSL-S (Houser et al., 2008); S362ANI (Kustowski et al., 2008); GyPSuM-S (Simmons et al., 2010); S40RTS (Ritsema et al., 2011); SAVANI (Auer et al., 2014); and SEMUCB-WM1  (French and Romanowicz, 2014). *TP* stands for "true positives" indicating a high-temperature cluster for mantle flow models and a low-velocity cluster for seismic tomographic models; similarly, *TN* ("true negatives") corresponds to a low-temperature cluster and a high-velocity cluster for flow models and seismic tomographic models, *FN* ("false negatives") corresponds to low-temperature cluster and low-velocity cluster for flow models and seismic tomographic models, respectively, and *A* is Earth's total surface area.

(iii)    The time-averaged median angular distance $\tilde{\theta}$ between volcanic eruptions and basal mantle structures. We considered three databases of volcanic eruptions: (1) large igneous provinces (LIPs) from Ernst and Youbi (2017), hereafter referred to as EY17. These LIPs are typically associated with plume heads (Richards et al., 1989), covering > $1e^5$ km$^2$ and erupted as igneous pulses 1-5 Myr long over less than 50 Myr  (Bryan and Ernst, 2008). EY17 contains 105 LIPs back to 960 Ma (Fig. 2). (2) LIPs from Johansson et al. (2018), hereafter referred to as J18. J18 contains 185 LIPs from 960 Ma (Fig. 2), which are compiled from Bryan and Ernst (2008), Coffin et al. (2006) and Ernst (2014). J18 also contains smaller oceanic islands and seamounts associated with plume tails (Coffin et al., 2006), which are longer-lived than transient plume heads (Richards et al., 1989). (3) Kimberlite eruptions from Tappe et al. (2018), hereafter referred to as T18.  T18 contains 983 kimberlite eruptions from 960 Ma (Fig. 2). We used *pyGPlates* Python API (www.gplates.org/pygplates) to identify the centroid of LIP polygons in databases EY17 and J18. We only considered LIPs covering an area greater than 20,000 km$^2$. We resampled databases EY17, J18 and T18 to obtain the same temporal resolution (20 Myr) as in the mantle flow models. Volcanic eruptions that occurred within $(a - a_w) < a \le (a + a_w)$, where $a_w = 10$ Myr is an age window, were assigned the age $a$. We combined 1168 centroids from EY17, J18 and T18 to measure the distance between volcanic eruptions and BMSs from 960 Ma. We created maps of angular distances from the edge of BMSs, with distances positive inside and negative outside BMSs (Davies et al., 2015b; Doubrovine et al., 2016; Flament et al., 2022). For a given tectonic reconstruction, minimum angular distances to BMSs were derived from these maps at reconstructed volcanic

eruption locations. These minimum distances were cumulated over the period of interest and represented as "sample" empirical distribution functions (EDFs). We reported the time-averaged median angular distance $\tilde{\theta}$ (with cumulative probability equal to 0.5) between volcanic eruptions and basal mantle structures. $\tilde{\theta} > 0$ indicates that 50% of volcanic eruptions are within BMSs, whereas $\tilde{\theta} < 0$ indicates that 50% of volcanic eruptions are outside BMSs. The absolute value of $\tilde{\theta}$ indicates the proximity of volcanic products to BMSs (in a median sense). The computation was carried out for mobile BMSs predicted by mantle flow models, and for stationary BMSs imaged by tomographic models.

(iv) The statistical significance of the median distance between volcanic eruptions and BMSs, expressed as a fraction $f_s$. For each mantle flow or tomographic model, we created 1000 distributions of spatially random points, each containing 1168 points with the same temporal distribution as in the database (Fig. 2). A set of 1000 "random" EDFs was created for each case. We computed the mean of absolute values of distances ($\overline{|\theta|}$) for each sample and random EDF. We carried out a Kolmogorov-Smirnov (KS) statistics test (Kolmogorov, 1933) based on the distance between two EDFs (the sample EDF and each random EDF). We used the scientific Python implementation of the KS test for two samples ([https://docs.scipy.org/doc/scipy/reference/generated/scipy.stats.ks_2samp.html](https://docs.scipy.org/doc/scipy/reference/generated/scipy.stats.ks_2samp.html)). In these tests, the null hypothesis that the sample EDF and each random EDF were accidentally selected from the same underlying population can be rejected if the KS statistic $D$ was greater than a critical value, a relationship which we expressed as (Knuth, 2014): $D_n > \sqrt{-\frac{\ln\left(\frac{\alpha}{2}\right)}{n}}$, where $n = 1168$ is the number of LIPs in both the sample and random EDFs, and $\alpha = 0.05$ is the confidence level. We computed the fraction $f_s$ of random EDFs that passed the KS test and for which $\overline{|\theta|}$ was greater than in the sample EDF. $f_s$ close to 1 indicate that the considered volcanic eruption locations are closer to the considered BMSs than random points (in a mean sense) and that the sample distribution is significantly different from the random distribution. In contrast, $f_s$ close to 0 indicate that random points are closer to the considered BMSs than the considered volcanic eruption locations (in a mean sense), and/or that the sample distribution is not significantly different from the random distribution.

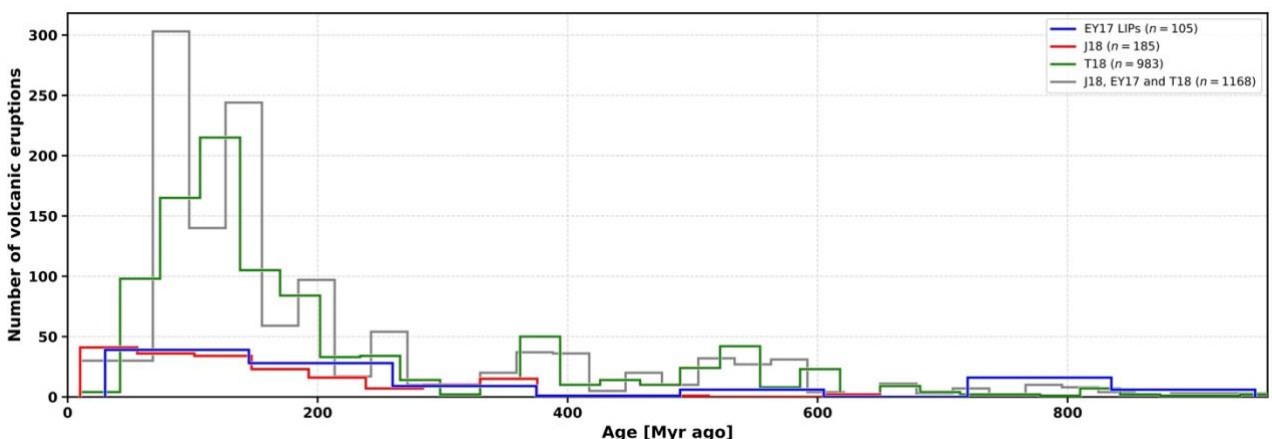

Figure 2. Number of volcanic eruptions as a function of age from 0 - 960 Ma for databases EY17 (Ernst and Youbi, 2017) J18 (Johansson et al., 2018) and T18 (Tappe et al., 2018). The number of volcanic eruptions from 960 Ma is given in brackets for each database, as well as the total number of volcanic eruptions.

## 2.4 Mantle structure cluster analysis

To facilitate an objective, quantitative comparison between 3D volumes of seismic velocity and model temperature fields in the lower mantle, we reduce these 3D volumes into 2D maps using vertical volume averaging (see Flament, 2019; Lekic et al., 2012). We use $k$-means clustering (Macqueen, 1967) to separately classify temperature and seismic velocity anomalies in the lower mantle into two groups. Profiles for each field are extracted beneath ~200,000 equally spaced points (with average distance 0.45º) at 31 depths between 1000 km and 2800 km (Lekic et al., 2012), and then separated into two clusters according to the variation in the property of interest with depth. We refer to these quantities as "lower mantle clusters".

In order to evaluate the models, as in Flament (2019), we compute the accuracy $Acc = (TP+TN)/A$ and sensitivity $S = TP/(TP+FN)$ to quantify the match between present-day lower mantle clusters from flow models and seven S-wave tomographic models: SAW24B16, Mégnin and Romanowicz (2000); HMSL-S, Houser et al. (2008); S362ANI, Kustowski et al. (2008); GyPSuM-S, Simmons et al. (2010); S40RTS, Ritsema et al. (2011); SAVANI, Auer et al. (2014); and SEMUCB-WM1, French and Romanowicz (2014). *TP* stands for "true positives" indicating that a high-temperature cluster for mantle flow models and a low-velocity cluster for seismic tomographic models; similarly, *TN* ("true negatives") corresponds to a low-temperature cluster and a high-velocity cluster for flow models and seismic tomographic models, *FN* ("false negatives") corresponds to low-temperature cluster and low-velocity cluster for flow models and seismic tomographic models, respectively, and *A* is Earth's total surface area.

## 3 Results

### 3.1 Implications of alternative reference frames

We compare five different reconstructions between 200 Ma and 900 Ma to assess the consequences of alternative assumptions and approach for reference frame construction (Fig. 3), including a standard paleomagnetic reference frame (Merdith et al., 2021) (PMAG), a no-net-rotation reference frame (NNR), our reference frame which is optimised with respect to tectonic rules (OPT), an orthoversion reference frame from Cao et al. (2021a) following Mitchell et al. (2012) (ORTHO) and a reference frame based on combination of paleomagnetic and geological data incorporating the alignment

of plume products at the surface with fixed LLSVP edges (Torsvik and Cocks, 2019) (FIX_LLSVP). At 200 Ma (Fig. 3a) all reconstructions are quite similar, however, by 300 Ma a visible difference is emerging between FIX_LLSVP and all other reconstructions in terms of the longitudinal positions of continents. At this time, South America is located about 20º farther westward in FIX_LLSVP compared with all other reconstructions, keeping in mind that there are only two tie points for a longitudinal correction based aligning plume products with the African LLSVP: the  Skagerrak Central LIP

and a Scandinavian kimberlite (Torsvik and Cocks, 2019). At 400 Ma the NNR, OPT and ORTHO reference frame are still very similar, but deviating substantially from both PMAG and FIX_LLSVP reference frames, in particular the longitudinal continental positions in the FIX_LLSVP frame, which ties kimberlites and LIPs in Siberia to the Pacific LLSVP (Torsvik and Cocks, 2019). This assumption requires the western edge of Laurussia (North America, Greenland and Baltica combined) to migrate ~80º westward back in time between 300 Ma and 400 Ma (Fig. 2a), translating to a

speed of ~9 cm/yr of a large continental mass including several cratons and a small amount of ocean crust. This is inconsistent with the analysis of Zahirovic et al. (2015) who found that the RMS speeds of plates with more than 50% of their area comprised of continental crust predominantly have RMS speeds between 2 cm/yr and 4 cm/yr. This reflects the substantial continental drag resisting plate motion in these cases, casting doubt on the viability of the FIX_LLSVP scenario. The differences between both OPT and NNR cases on the one hand with FIX_LLSVP on the other hand are

even more extreme at 500 Ma. In FIX_LLSVP Laurentia (North America and Greenland) is aligned with the Pacific LLSVP, ~45º farther west than at 400 Ma (Fig. 3a), while the longitudinal positions of Laurentia and Gondwana have not changed much in the OPT and NNR reconstructions, reflecting both the minimisation of net rotation as well as limits imposed on the speeds of continents in the OPT case.  The PMAG reconstruction is quite different to all other models, as expected, as it is not a mantle reference frame and implies no constraints on longitude.


        For times older than 500 Ma (Fig. 3b) we only compare four reconstructions as the model by Torsvik and Cocks (2019) does not reach back to 600 Ma.  Firstly, there is distinctive similarity between the OPT and NNR reconstructions, even though the OPT case minimises trench migration speeds and penalises global continental velocities in addition to minimising net rotation. The primary reason for this is that trench migration and net rotation are relatively closely coupled,

so minimising one also reduces the other to a large extent, and continents typically do not tend to move fast when both

net rotation and trench migration is minimised, even if explicit constraints are not introduced for continents. This comparison provides an important insight, namely that the simple lithospheric no-net-rotation rule used to produce the NNR model produces results that are not dramatically different from a model optimised by a set of more general tectonic rules. This is important because NNR models have been frequently used in tectonic and mantle flow models for practical

reasons (e.g., Mao and Zhong, 2021; Zhong and Rudolph, 2015; Behn et al., 2004; Kreemer and Holt, 2001) in the absence of other available mantle reference frames. Our results suggest here that NNR reference frames are not entirely unrealistic from a tectonic rules point of view. In terms of the relative importance of plate motion optimisation parameters, our results suggest that minimising net rotation is the most important one, with minimising subduction zone migration of secondary importance, as minimising net rotation also reduces subduction zone migration to some extent (also see Müller

et al. (2019) for a discussion of the effect of changing the relative weight of these parameters). Preventing the speed of continents to exceed continental speed limits is the least important parameter. We introduced it to ensure that large swathes of synthetically reconstructed ocean floor would not result in a minimal net rotation solution that imposes unreasonable motions on the smaller continental regions.

The PMAG model is expectedly quite different, with no longitudinal and plate speed constraints imposed, while

the orthoversion model (ORTHO) is very different as well by design, as it follows the idea that Rodinia formed about 90º east from Pangea, such that Pangea assembled over the western subduction girdle bounding Rodinia (Mitchell et al., 2012). Interestingly, our OPT reconstruction implies instead that Pangea formed about 90º east of Rodinia. We stress that this model behaviour emerges naturally from our optimisation parameters without being imposed. It is important to note that Mitchell et al. (2012) found that their data could not discriminate well between one or the other scenario, i.e. Pangea

either forming 90º east or west of Rodina. They merely favoured a westward migration of the two successive supercontinent centres because they thought this solution would minimise plate speeds. Instead, our optimised reconstruction suggests that the opposite solution minimises plate velocities. In summary, an unexpected outcome of our optimised absolute reference frame is that it generates an orthoversion-consistent reconstruction by following a completely different approach from that used by Mitchell et al. (2012), thus independently lending support to the concept

as a natural evolutionary pathway of successive supercontinents.

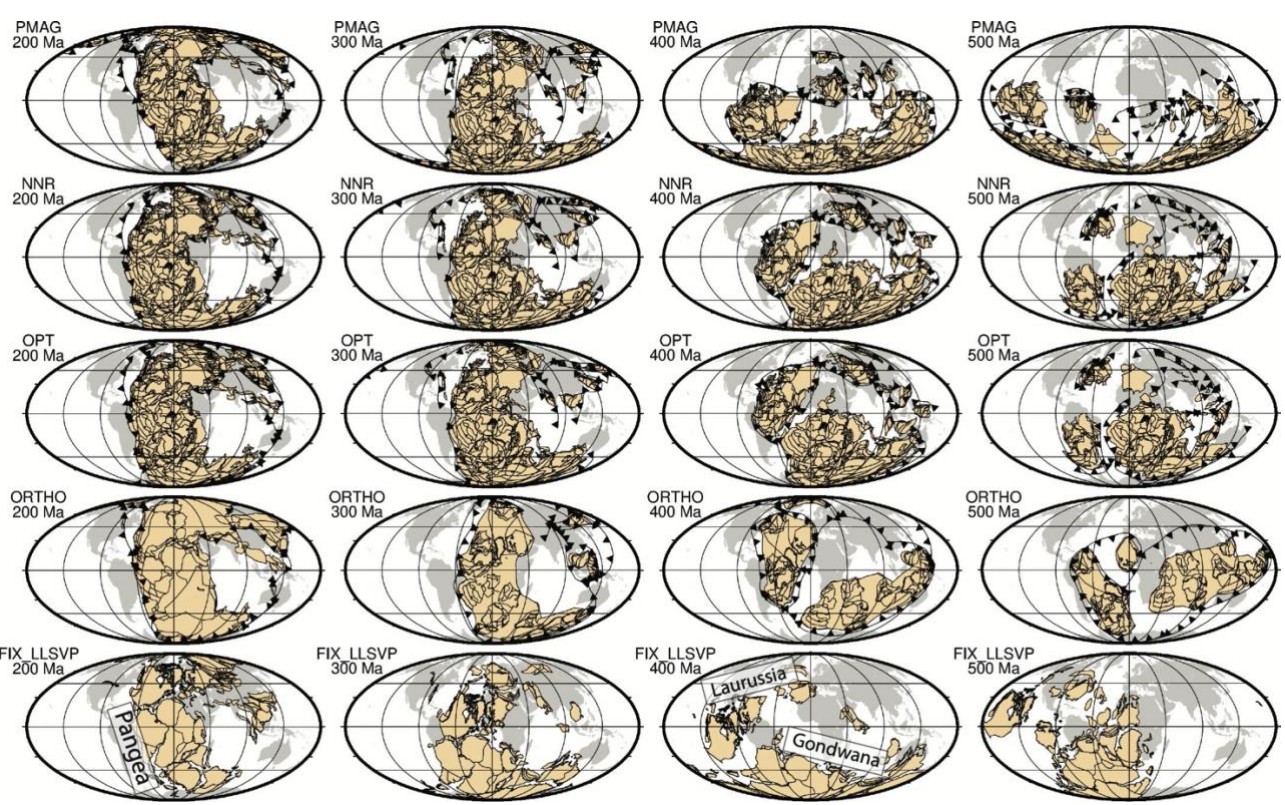

Figure 3. Plate reconstructions based on alternative approaches for modelling absolute plate motions, with reference frames based on paleomagnetic data (PMAG) (Merdith et al., 2021), no-net-rotation (NNR), tectonic rules-based optimisation (OPT), orthoversion from Cao et al. (2021a) following Mitchell et al. (2012) and a combination of paleomagnetic data with aligning LIPs with the edges of LLSVPs assumed to be stationary (Torsvik and Cocks, 2019), covering the time period from 200-900 Ma. The reconstruction of Torsvik and Cocks (2019) does not extend back to 600 Ma and older. Continents are outlined in beige while subduction zones are toothed black lines. The present-day position of continents is shown in light grey in the background as a reference.


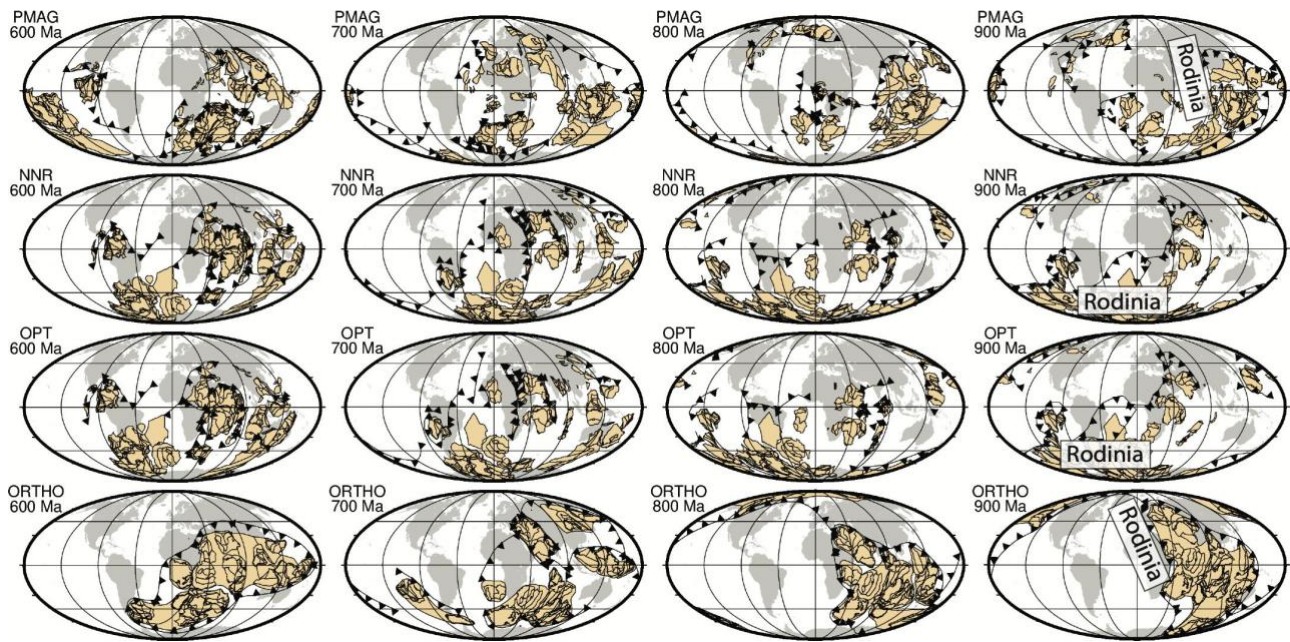

Figure 3 (*continued*)

## 3.2 Lithospheric net rotation

The dramatic reduction of lithospheric net rotation in our optimised reference frame relative to the PMAG reference frame is illustrated in Figure 4a, which compares net rotation of our optimised model with the paleomagnetic reference frame from Merdith et al. (2021) and the mantle reference frame model by Matthews et al. (2016) incorporating the Paleozoic reconstruction from Domeier and Torsvik (2014), with the latter being similar to the model of Torsvik and Cocks (2019) shown on Figure 3. These two Paleozoic models (Domeier and Torsvik, 2014; Torsvik and Cocks, 2019)

are constructed as mantle reference frames, by following the idea that by applying a TPW correction and aligning LIPs and kimberlites with the edges of LLSVPs an approximation of the "true" latitude and longitude of plates relative to the mantle is obtained. If this were the case, we would expect to see the large fluctuations in lithospheric net rotation seen in a model based on paleomagnetic data alone (Fig. 4b) dramatically reduced. However, the result of applying empirical TPW and longitudinal corrections as proposed by Domeier and Torsvik (2014) results in 0.4°-1.5°/Myr of net lithospheric

rotation, which is up to five times larger than the rates regarded as reasonable based on geodynamic considerations (Becker, 2006; Conrad and Behn, 2010) (Fig. 4b). In contrast, if absolute plate motions are jointly optimised for minimising net rotation, trench migration and fast continent velocity, we obtain a model that displays net rotation with rates less than 0.25°/Myr.

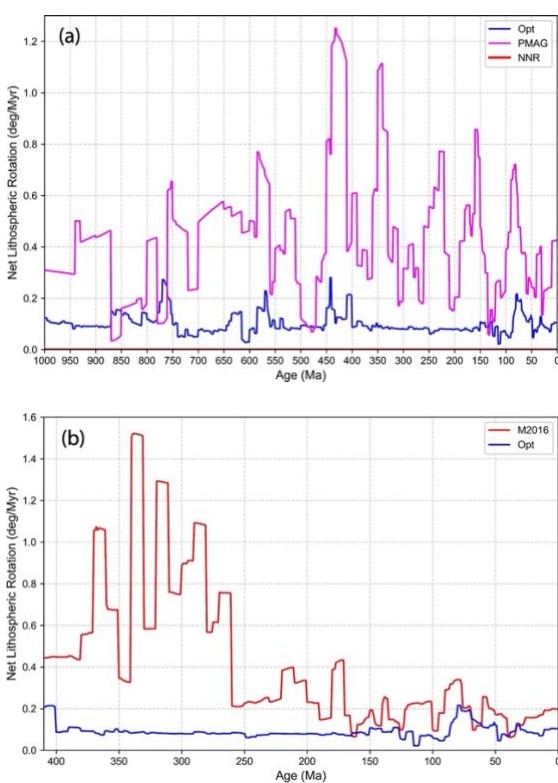

Figure 4. (a) Net lithospheric rotation of the paleomagnetic reference frame from Merdith et al. (2021) (PMAG, magenta) versus our optimised reference frame (OPT, blue) with weights for NR=1 (net rotation), TM=0.5 (trench migration), and PV=0.5 (absolute plate velocity of continental regions) from 1000 Ma to the present with the NNR model shown for reference in red with zero net rotation. For the period of 80-0 Ma hotspot tracks are used in addition as part of the optimisation, with a weight HS =1 (fitting of present-day hotspots to age-progressive seamount tracks). Net rotation is below 0.25°/Myr, as recommended by independent geodynamic studies (Conrad and Behn, 2010; Becker, 2006). (b) Net rotation of the mantle reference frame model by Matthews et al. (2016) which is based on a combination of the models by Müller et al. (2016) from 230-0 Ma and Domeier and Torsvik (2014) for the period of 410-250 Ma, compared to our optimised model. The Domeier and Torsvik (2014) model is similar to the model by Torsvik and Cocks (2019) (Fig. 2), but the latter does not include plate topologies, and can therefore not be used to compute net rotation.

### 3.3 Subduction zone migration

Compared to the paleomagnetic model (Fig. 5a), our optimised model (Fig. 5b) exhibits significantly reduced trench-orthogonal subduction zone migration scatter as well as reduced median absolute deviation of trench motion (Fig. 6), reflecting the suppression of fast, geodynamically unreasonable global trench migration rates. The substantial overall

improvement in the scatter of trench migration velocities is expressed in limiting the bulk of trench advance to a relatively narrow band of rates to 0-3 cm/yr (Figs 5a, b). Subduction zone retreat exhibits more scatter, particularly between 150 Ma and 100 Ma and before 190 Ma, but the majority of retreating trench speeds are confined to 0-4 cm/yr for most of the model. In addition, the optimised model exhibits a notable improvement over the no-net-rotation model (Fig. 5c) in confining the majority of trench advance to 0-2 cm/year and the majority of trench retreat to 0-4 cm/year (c). There are periods during which the scatter of trench migration speeds is systematically smaller or larger than those observed during Pangea stability and after Pangea breakup (Fig. 5b). In the last 320 million years the Median Absolute Deviation (MAD) of subduction zone migration is largely between -1 cm/yr (trench advance) and 2 cm/yr (trench retreat), with the exception of the Early Cretaceous period with a MAD of up to 4 cm/yr (Fig. 6). During periods of Rodinia amalgamation (prior to 940 Ma) and stability (940-870 Ma) the MAD of trench migration is confined to -1/+2 cm/yr, while Pangea assembly/stability from 320 Ma to ~200 Ma is characterised by a range of -2/+2 cm/yr (Fig. 6). The relative subduction zone stability during these periods reflects that during the late stage of closure of internal ocean basins that are consumed in the process of supercontinent amalgamation, subduction zone migration slows, and during phases of stability of a large continental mass the ring of subduction zones surrounding it is also relatively stable.

The opposite holds for times of supercontinental dispersal, during which the ring of subduction zones surrounding a supercontinent rolls back oceanward, accommodating the creation of new internal ocean basins. The spread of trench migration during the dispersal of Rodinia (~870-650 Ma) is in the range of -2/+5 cm/yr (MAD, Fig. 6), slightly larger than that observed during the last 200 Ma, but the difference may simply be an artefact of much larger uncertainties for Neoproterozoic plate reconstructions. The period that stands out by displaying by far the largest scatter of trench migration, with the bulk between -3/+6 cm/yr, with some even larger outliers is the period from 600 Ma to 320 Ma (Fig. 6). This period is the most dynamic time in terms of subduction zone migration in the last billion years and comprises most of the Paleozoic era with the exception of the Late Carboniferous and Permian. This is the antithesis of the "boring billion" (Brasier, 2012) – we propose to call it the "zippy tricentenary era", in short the "zippy tricentenary". Multiple internal ocean basins were destroyed in the process of the formation of Gondwana between 600 Ma and 550 Ma, while after 490 Ma the ephemeral Iapetus Ocean was replaced by the Rheic Ocean, separating several arc terranes from northern Gondwana.

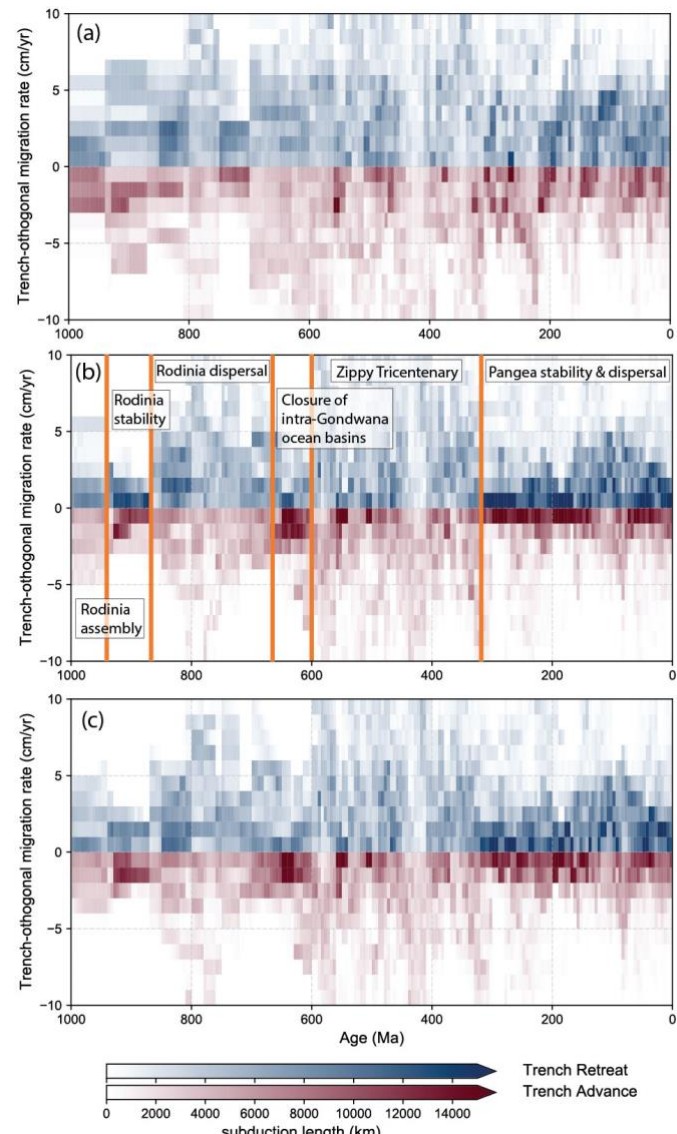

Figure 5. Histogram of the trench-orthogonal overriding plate speed for (a) the plate motion model from Merdith et al. (2021), (b) the same model with net lithospheric rotation removed and (c) our optimized mantle reference frame. Colours are proportional to the length of subduction zones which are either retreating (blue) versus advancing (red) at a given

rate. Both the no-net-rotation and optimised models limit the occurrence of unreasonably fast trench retreat or advance.

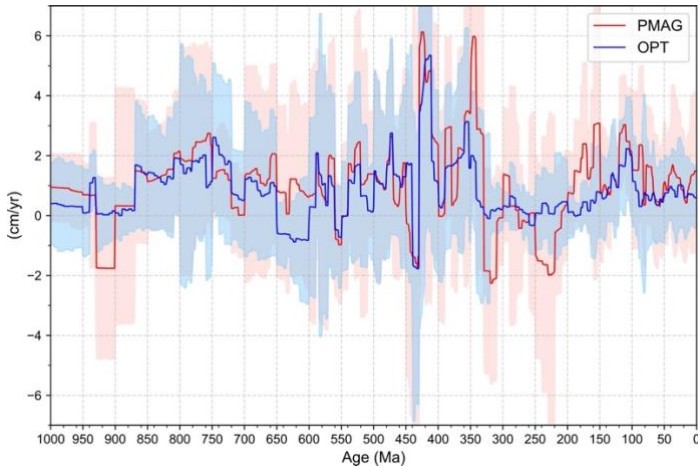

Figure 6. Median trench motion speed with median absolute deviation error range for the unoptimized plate motion model in red and the optimised model in blue, using a 5 Myr moving average window. Periods of supercontinent stability are characterised by very slowly moving subduction zones, but Rodinia dispersal and the following long period of the successive opening and closing of a number of internal ocean basins resulted in a larger prevalence of relatively fast subduction zone migration compared with Pangea dispersal. See text for discussion.

### 3.4 Continental and plate speeds

The third parameter we use to impose tectonic rules on our optimised plate motion model is global continental RMS speeds. The paleomagnetic reference model (Fig. 7a) is characterised by plate and continental speeds that are frequently 50% or more above those of the optimized model (Fig. 7b), which both limits maximum continental RMS speeds below the continental "speed limit" of 10 cm/yr (Zahirovic et al., 2015) whilst minimizing peaks in plate speeds. The short-lived peak in plate speeds at 590-570 Ma is likely related to an artefact in the reconstruction of now subducted ocean basins.

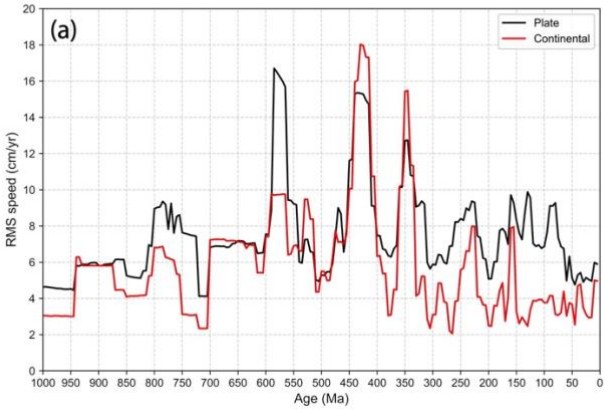

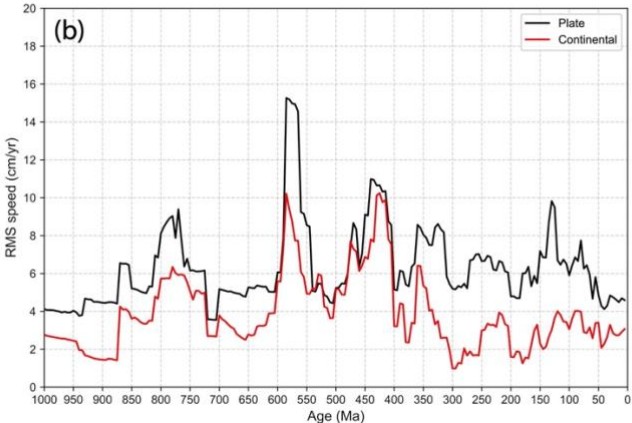

Figure 7. Root mean square (RMS) speeds of all plates and continents in the paleomagnetic model (a), and the optimised model (b). The paleomagnetic model displays large RMS speeds of plates and continents up to 18 cm/year. The no-net-rotation model alleviates outliers to a large extent, with the optimised model especially reducing mean continental speeds further. Some time periods, notably 600-550 Ma and 450-400 Ma, still show relatively large RMS speeds, likely reflecting artefacts in the reconstruction of relative plate motions and plate boundaries. Such artefacts mostly occur due to the way synthetic, now subducted ocean crust is reconstructed. These reconstructions assimilate preserved geological evidence related to the types of regional plate boundaries and the timing of the opening or closing of ocean basins, but are nevertheless subject to interpretation, and seafloor spreading rates are only inferred based on ensuring self-consistent relative plate kinematics such as divergence between continents or convergence at subduction zones margins. As a consequence, RMS speed artefacts can arise, which can be addressed in future, improved plate and plate boundary reconstructions.

### 3.5 Mantle flow models

Our mantle flow models are driven by imposed surface plate velocities, subduction zone locations and geometries (Supp. Animation S1) and reconstructed age-area distributions of the ocean crust through time (Figs 8, 9, Supp. Animation S2), following the method of Williams et al. (2021). The predicted evolution of mantle temperature primarily records the effect of changing subduction zone topologies and convergence velocities and the age of subducting slabs through time. Here we focus on the evolution of basal mantle structure (Figs 10 and 11), particularly relevant for

understanding the history of deep mantle plumes, and on the upper mantle structure through time which is connected with surface magmatism via upper mantle temperature anomalies and upwellings. To do this, we compare output from two mantle flow models, OPT1 (Fig. 10a) and OPT2 (Fig. 10b), which differ in buoyancy ratio for the basal mantle layer (Table 1) (see also Supp. Animations S3-S10). Model outputs for the NNR and PMAG models are included in the supplementary material (Figs S1-S4, S6 and S7).

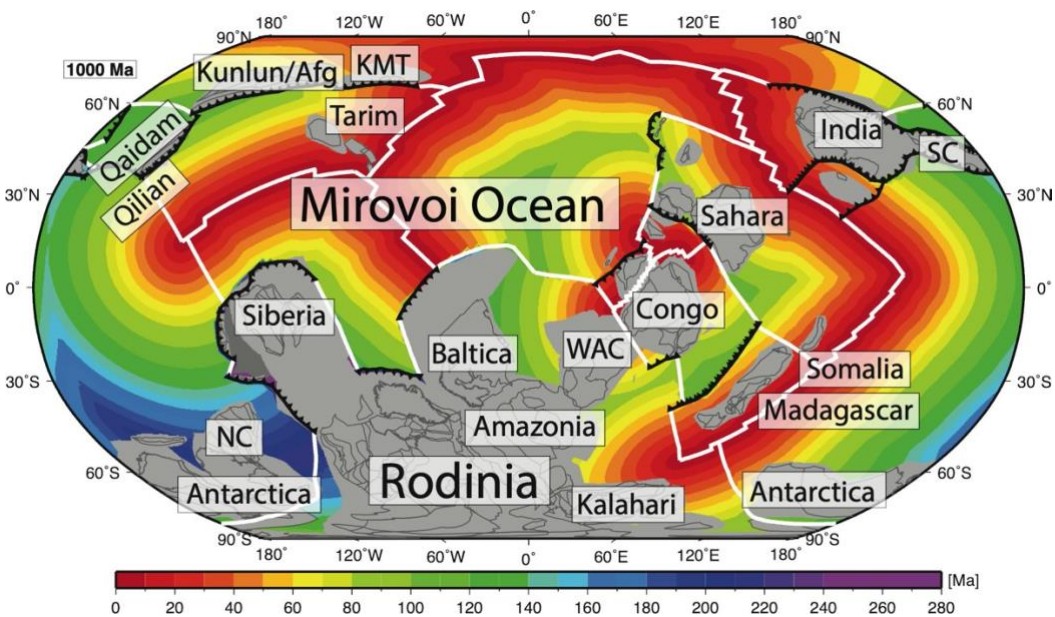

Figure 8. Reconstruction at 1 Ga with synthetic age of the ocean floor reconstructed using the method by Williams et al. (2021) with mid-ocean ridges shown as white lines, subduction zones as toothed black lines, regions of continental crust filled with grey and individual continental blocks are labelled; Afg = Afghanistan, KMT = Kyrgyz Middle Tianshan, SC

= South China, NC = North China, WAC = West Africa Craton.

      During the first 400 million years of model evolution (1000-600 Ma) the basal mantle structure is dissected into a network of ridges as a response to a widely dispersed network of subduction zones from 1000-760 Ma, preventing any extensive basal mantle structure akin to present-day LLSVPs to form (Fig. 10a). Between 760 Ma and 560 Ma, an

equatorially-centred subduction girdle forms in our model, restricted to a latitude range less than 60° (Fig. 10a), accompanied by an arrangement of the continents within the same latitudinal belt. This gives rise to the formation of two coherent, extensive polar basal mantle structures, connected by a small number of evolving, ephemeral basal ridges, which persist to 500 Ma (Fig. 10a). The subsequent movement of Gondwana, some fragments of Eurasia and associated subduction zones to higher latitudes starts dissecting the previously formed polar basal mantle structures while a coherent Pacific basal structure emerges around 400 Ma (Fig. 10a). As no subduction zones migrate into this region after 400 Ma, the structure consolidates itself over the next 200 million years, without any equivalent on the opposite, African side forming, reflective of the persistent subduction zones between Gondwana and Laurussia (Figs. 9, 10, 11). A coherent sub-African basal mantle structure only starts emerging in our model about 20-40 million years after the breakup of Pangea, reflecting the time involved in the effect of sinking slabs on the lowermost mantle structure to fade away after cessation of subduction between Gondwana and North America after 380 Ma (Figs 9, 10, 11).

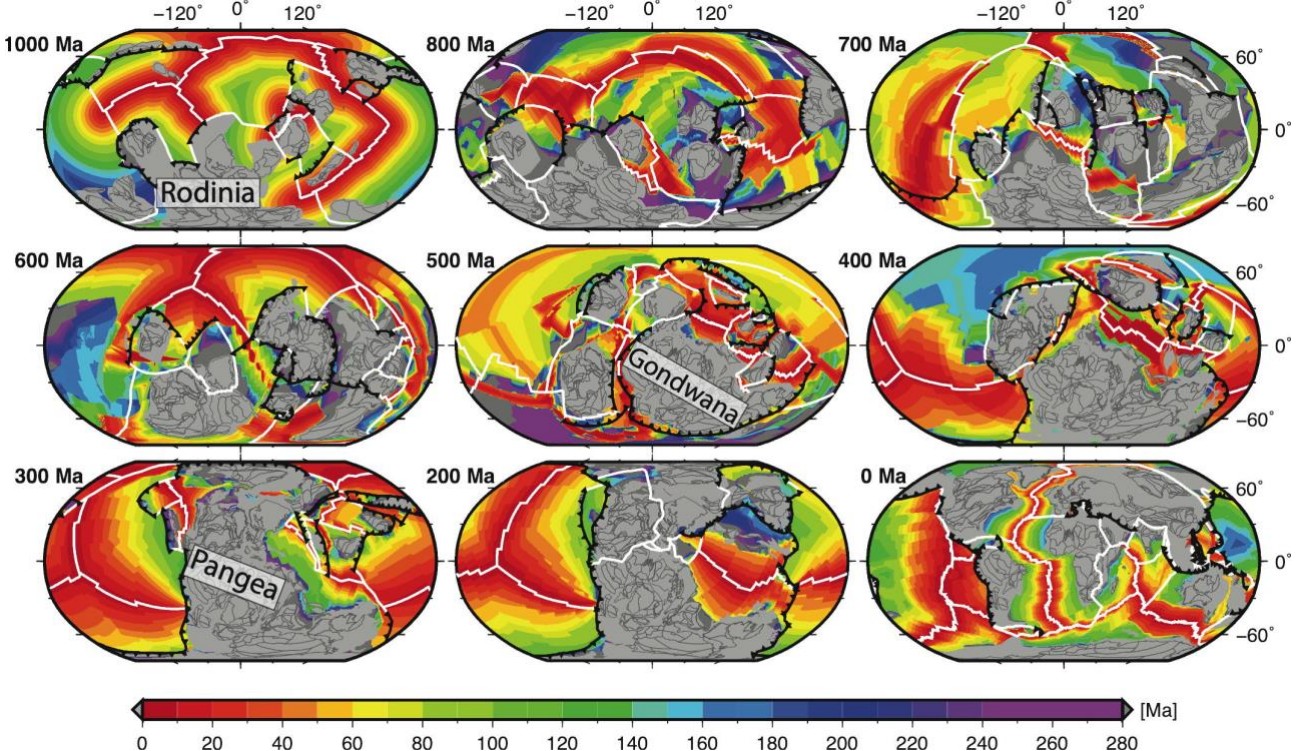

Figure 9. Oceanic crustal age grids from 1 Ga to the present constructed from plate rotations and boundaries in 100 Myr intervals following Williams et al. (2021), with plate boundaries and continents coloured as in Fig. 8.

These results are consistent with the inference from Cao et al. (2021a) that it may take 160-240 Myr for the basal mantle structure to reflect changes in subduction zone geometry at the surface. Our model thus records five distinct intervals of mantle convection geometry: (1) a network of dissected basal ridges (1000-600 Ma), (2) a short-lived degree-

2 basal mantle structure with upwellings centred on the north and south pole (600-500 Ma), (3) a transitional state in which the north polar basal structure migrates southward and gradually evolves into a Pacific-centred structure while the
575 south polar structure is dissected by subducting slabs and disintegrates into a network of ridges (500-400 Ma), (4) a Pacific-centred degree-1 structure (400-200 Ma) and (5) a degree-2 structure akin to what is observed today (160-0 Ma) which is composed of a long-lived Pacific basal structure joined by an African counterpart which gradually amalgamates during a ~40 Myr transition after the breakup of Pangea at 200 Ma. The overall evolution described above is similar for models OPT1 (Figs 10a, 11a) and OPT2 (Figs 10b and 11b), suggesting that the excess density of the basal mantle layer
may play a secondary role, in comparison to the imposed plate motion history, in driving large-scale mantle structure through time (Figs 10a and b, Supp Figs S1-S4, Supp. Animations S3-S6). We show how basal mantle structures under Africa, the Pacific, and elsewhere (in-between) have evolved through time based on model OPT1 (Fig. 12a). The predicted horizontally-averaged temperature and viscosity of model OPT1 at 1000 Ma and at present-day (Fig. 12b) illustrate that the model mantle temperature profile does not change significantly during the entire model run and that the
mantle viscosity profile follows that of Steinberger and Calderwood (2006).

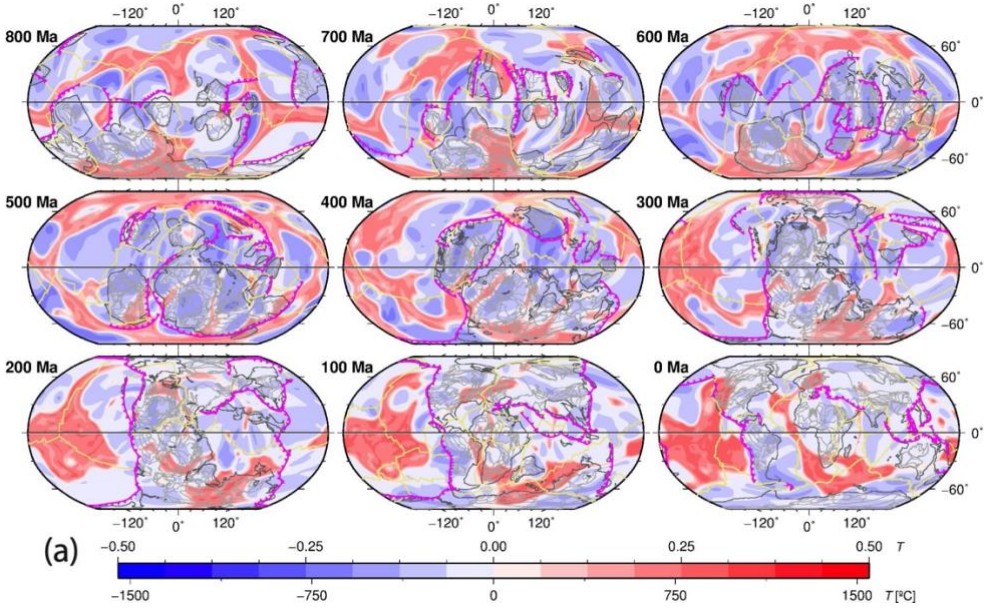

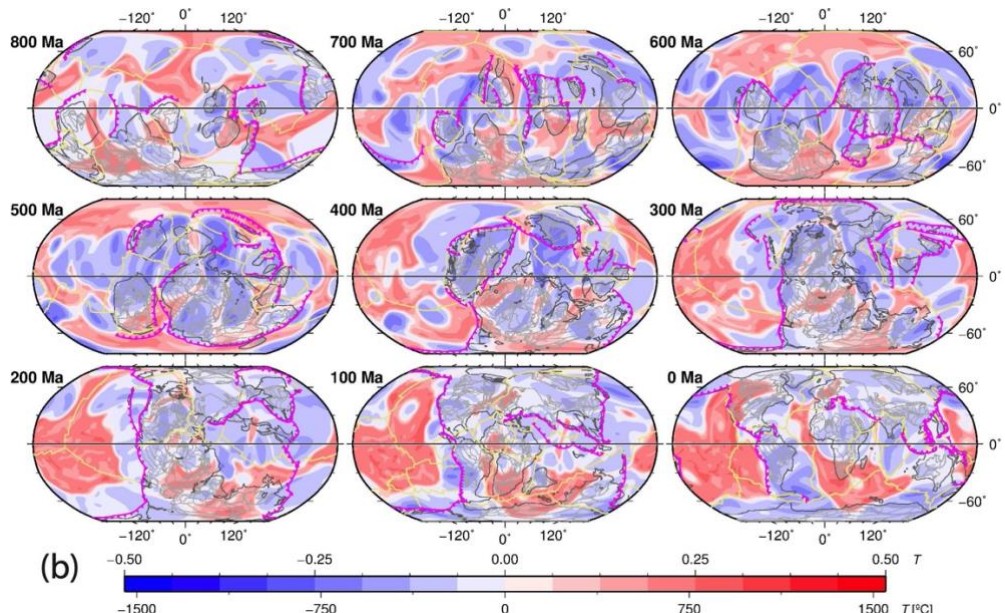

Figure 10. Map view of mantle temperature anomalies relative to the mean temperature at 2677 km depth for our geodynamic reference model OPT1 (a) and model OPT2 (b) from 800 Ma to the present (see Table 1 for model parameters). Reconstructed present-day coastlines and continental sutures are shown as thin grey lines, while outlines of continents are displayed as bold grey lines. Subduction zones are bold magenta lines with triangles pointing towards overriding plates while mid-ocean ridges are shown as yellow lines. The black line along the equator highlights the location of the mantle cross-sections shown in Figure 8.

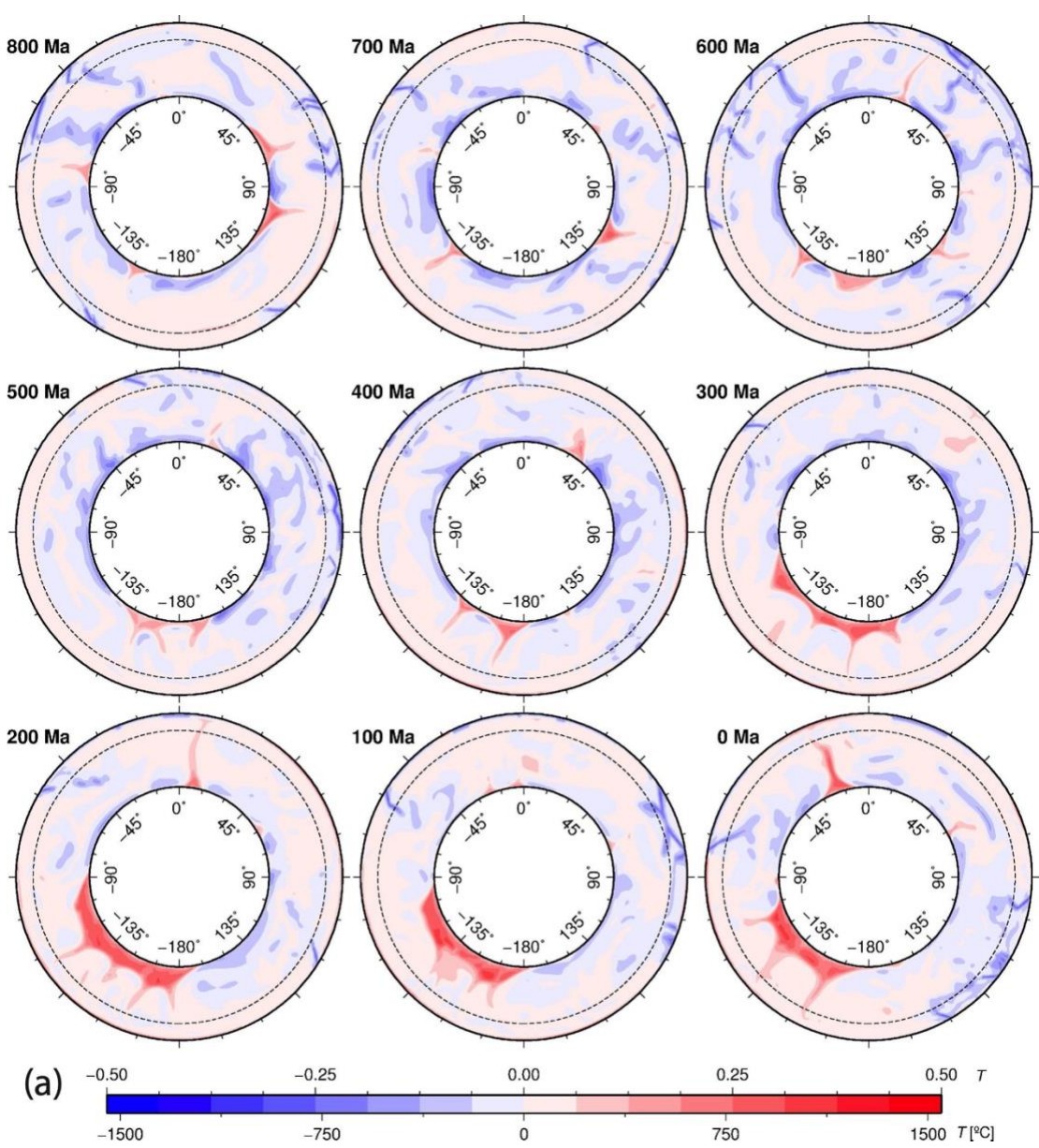

(a)

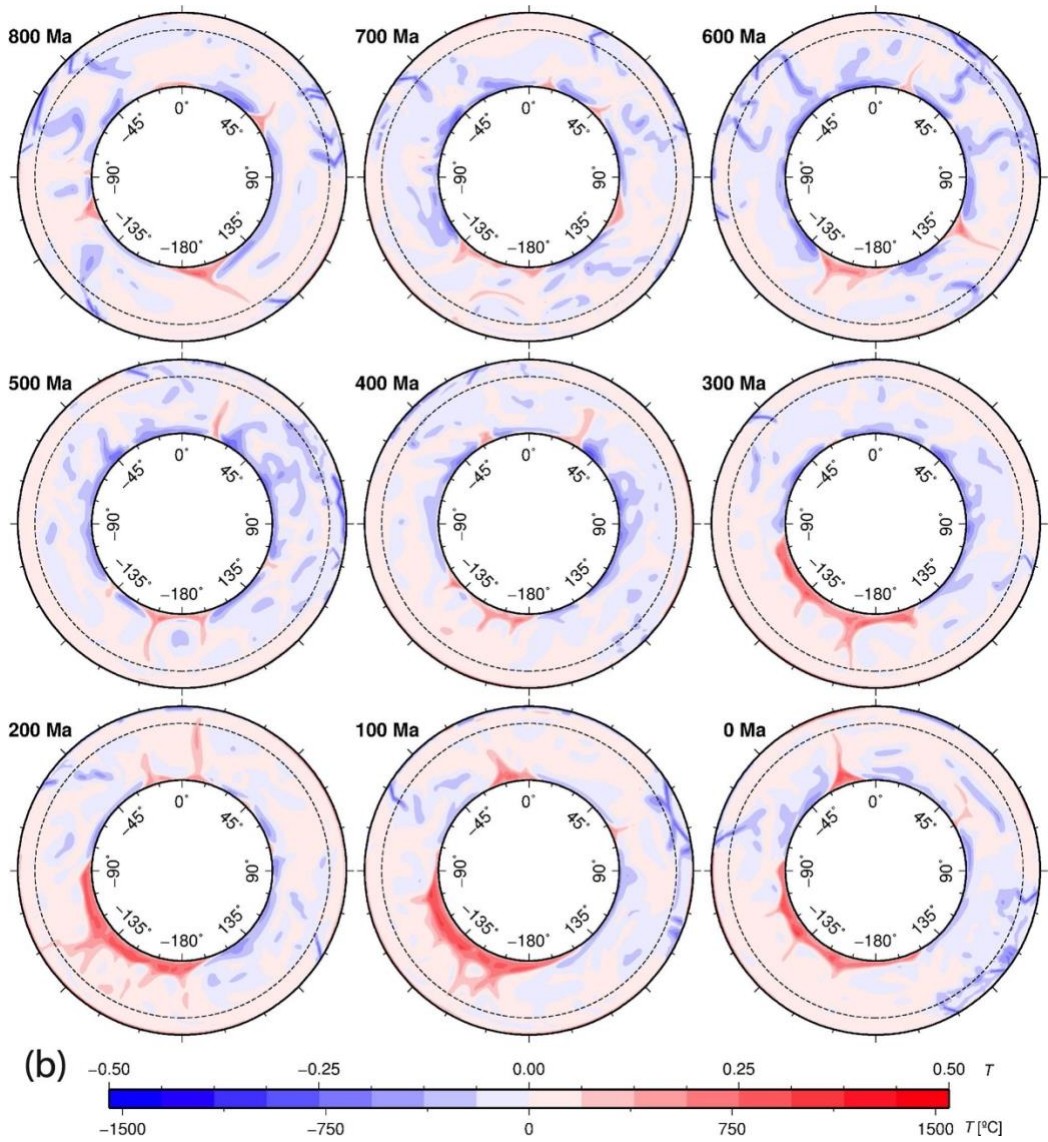

Figure 11. Global equatorial mantle cross-sections for our geodynamic reference model OPT1 (a) and OPT2 (b), distinguished by a difference in the excess density of the basal mantle material (see Table 1) in 100 Myr increments since 800 Ma. The dashed black line is the boundary between the upper and lower mantle. Numbers above the colour palette

represent non-dimensional temperature anomalies, while numbers below the colour palette are dimensional temperature anomalies.

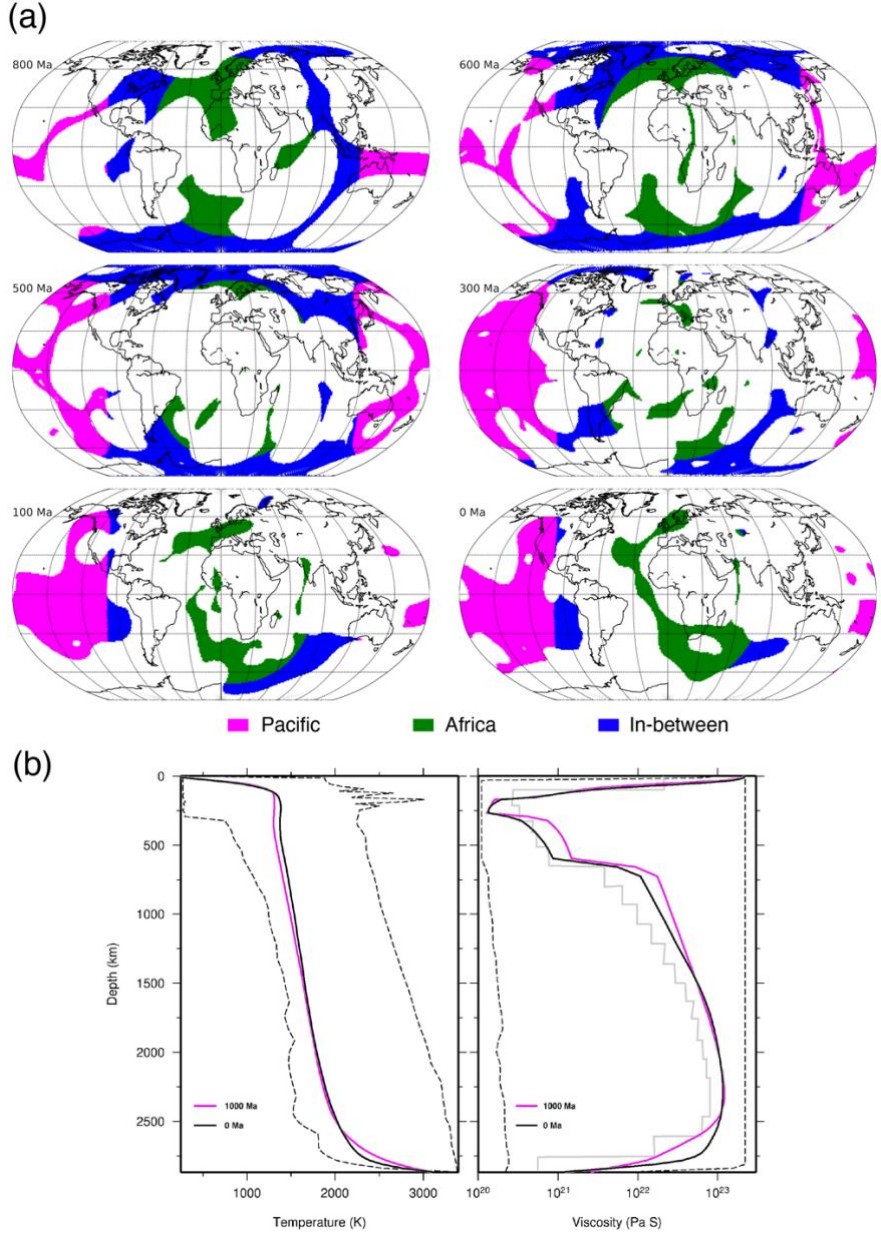

Figure 12. (a) The basal mantle structures at 2677 km depth for model OPT1 separated into three equal-area parts from 800 Ma to present-day. Basal mantle structures below present-day Africa (centred on lon/lat=11ºE/0ºN, with a radius of 70.5 º) are green, structures below the Pacific Ocean (centred on lon/lat=169ºW/0ºN, with a radius of 70.5º) are magenta, and structures elsewhere are blue; (b) predicted horizontally-averaged temperature (left panel) and viscosity (right panel) at 1000 Ma and 0 Ma for mantle flow model OPT1. The black dashed lines show the minimum and maximum values at the present-day in each panel. The grey line in the right panel shows the viscosity profile estimated by Steinberger and Calderwood (2006).

Upper mantle temperature anomaly maps at ~400 km depth (Fig. 13, Figs S5-S7) complement the view of the lower mantle evolution of our model and allows us to evaluate the mantle temperature response to the history of subduction as well as upwellings underneath continents, which is of interest as some continental magmatism may be partly driven by upper mantle temperature anomalies (as well as compositional anomalies but these are not modelled here). These maps (Fig. 13) highlight time periods during which continental regions overrode subducting slabs, leading to cooler than average upper mantle temperatures in these regions, while at the same time potentially enriching the mantle transition zone in these regions with volatiles from subducting slabs (e.g. Mather et al., 2020; Safonova et al., 2015; Cao et al., 2021c). This is observed under Siberia, Baltica and North America between 420-380 Ma, under North America between 100-40 Ma and along the rim of eastern/southern Asia and Zealandia after 100 Ma (Fig. 13) (see also Supp. Animations S7-S10).

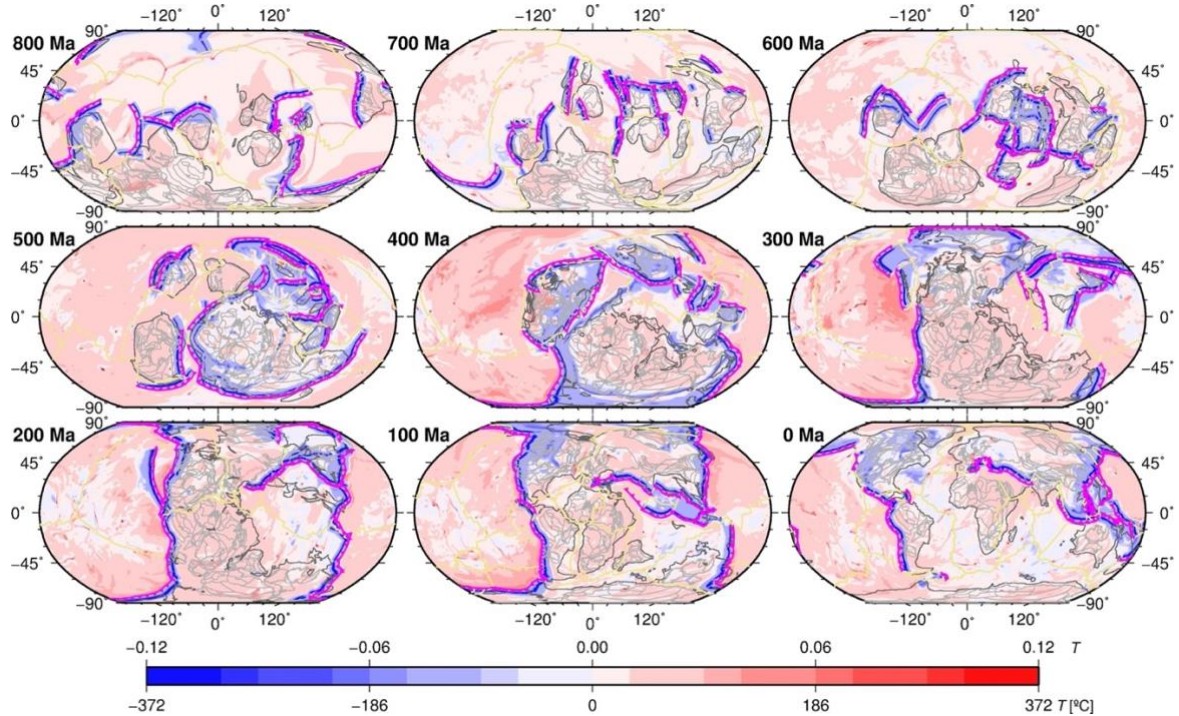

Figure 13. Map view of mantle temperature anomalies relative to the mean temperature at 396 km depth for our geodynamic reference model OPT1 in 100 Myr increments since 800 Ma. Reconstructed present-day coastlines and continental sutures are shown as thin grey lines, while outlines of continents are displayed as bold grey lines. Subduction zones are bold magenta lines with triangles pointing towards overriding plates while mid-ocean ridges are shown as yellow lines.

We use virtual transparent globes displaying the modelled time-dependent mantle temperature structure to

640 visualise the response of the 3D geometry of the basal layer and associated upwellings to the evolving geometry and

volume of slabs descending in the mantle (Fig. 14). The development of basal mantle structures in response to subduction

in our model is illustrated in two views of the mantle through time: One view is centred at 270ºE, a meridian crossing the

centre of Rodinia in the south and the Mirovoi Ocean in the north at 800 Ma while straddling the eastern Pacific Ocean

at present-day (Fig. 14a) while a second view is centred at 150º E, straddling subduction zones bounding the eastern

portion of Rodinia in the south and crossing the boundary between India and the Mirovoi Ocean in the north (Figs 8, 9

and Supp. Animation S11).

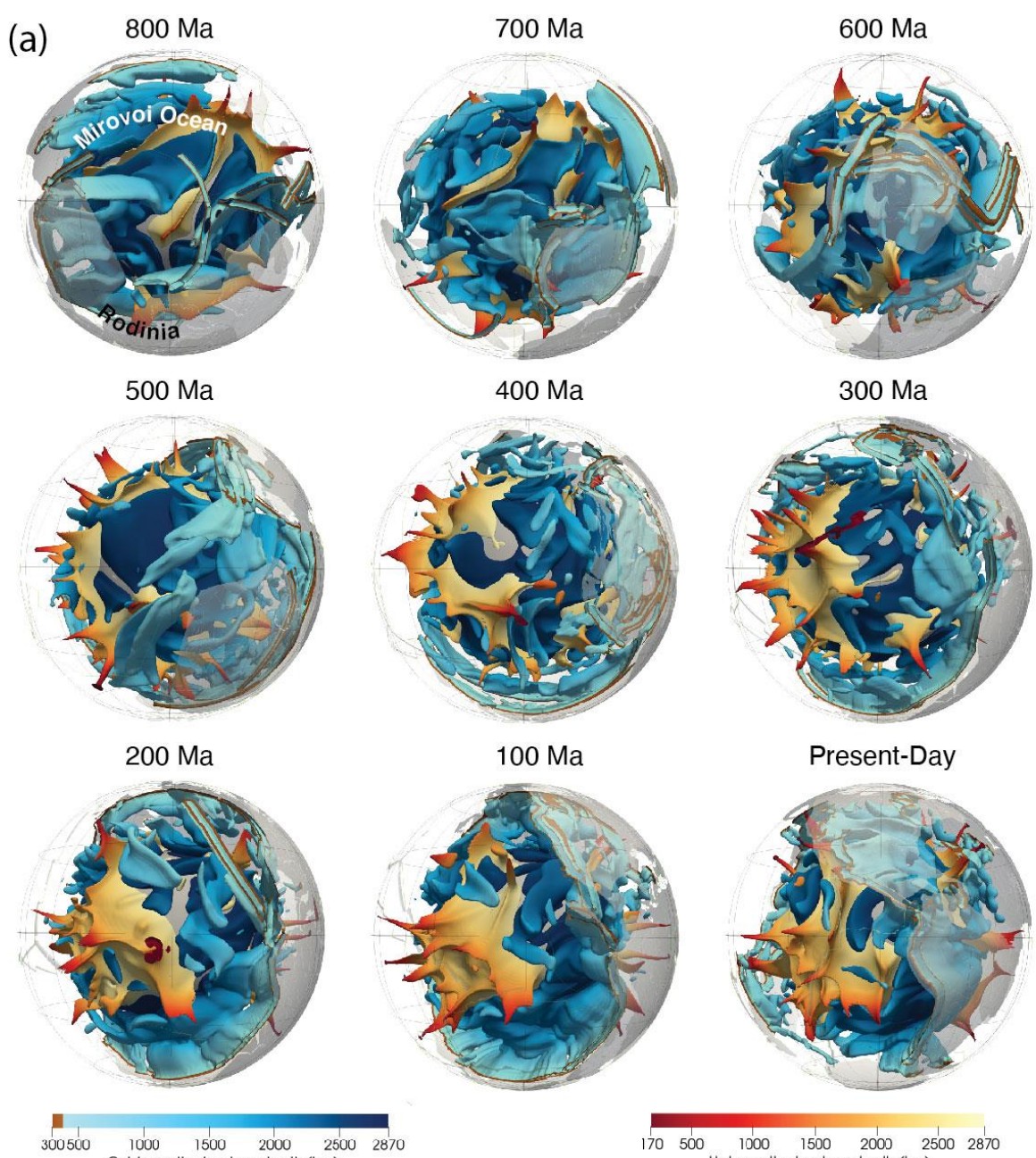

(a)

800 Ma    700 Ma    600 Ma

Mirovoi Ocean

Rodinia

500 Ma    400 Ma    300 Ma

200 Ma    100 Ma    Present-Day

300 500   1000   1500   2000   2500   2870
Cold mantle structure depth (km)

170  500   1000   1500   2000   2500   2870
Hot mantle structure depth (km)


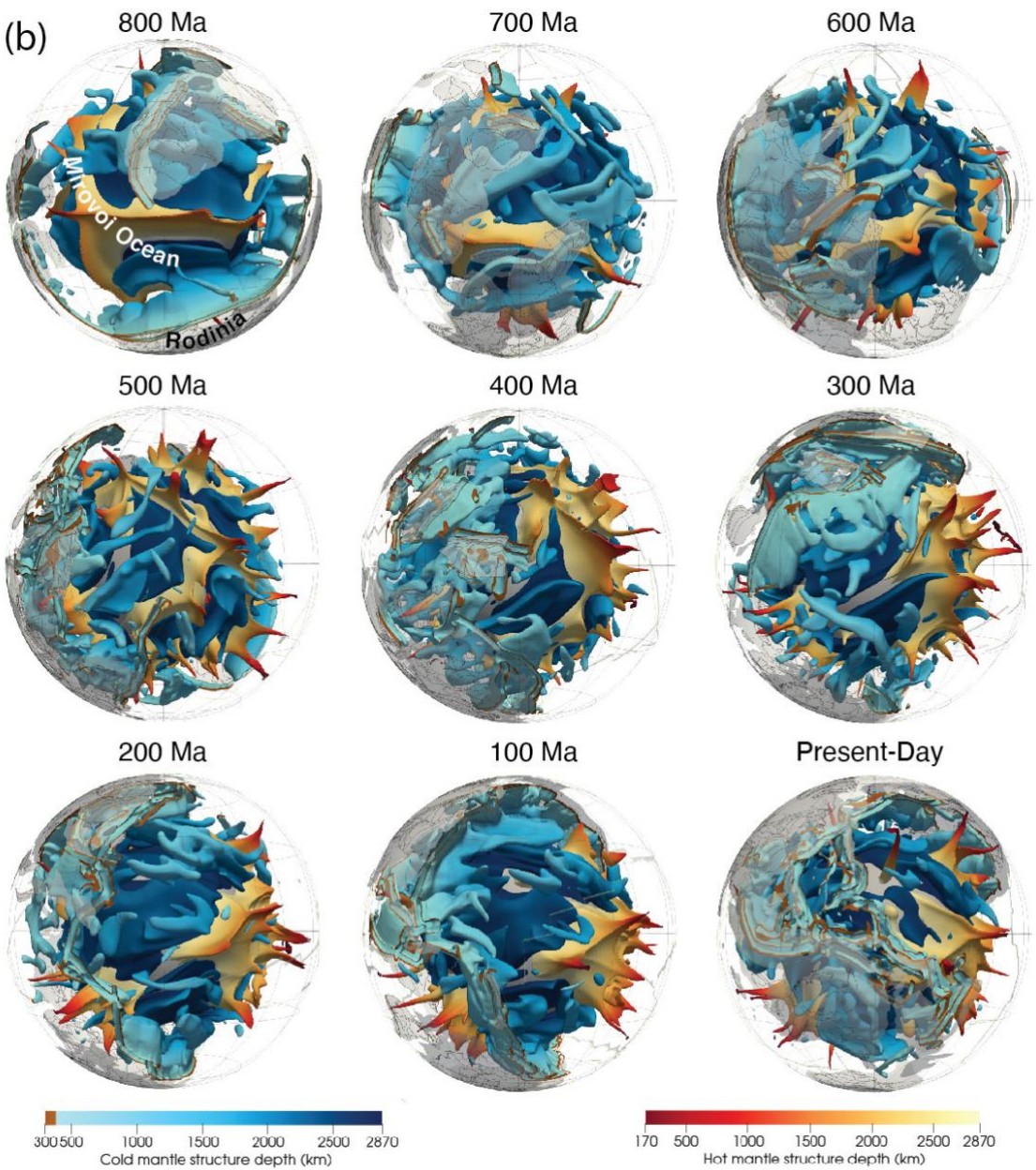

Figure 14. Visualisation of modelled mantle structure through time focussing on the Pacific hemisphere since 800 Ma in 100 Myr intervals with central meridians at 270°E (a) and 150°E (b). Mantle hotter than the layer average by non-dimensional value 0.1 (305 K) is shown in orange while mantle colder than the layer average by non-dimensional value -0.05 (153 K) is shown in blue, highlighting anomalously hot and cold mantle structures, largely corresponding to upwellings and subducting slabs, respectively.

### 3.6 Match of large igneous provinces and kimberlites with basal mantle structure through time

As noted in previous work (Torsvik et al., 2010), the volcanic eruptions reconstructed at their time of eruption and shown at present-day are generally close to LLSVPs, although this depends on the reconstruction that is used (Fig. 15). Reconstructed locations of volcanic eruptions are also close to mobile BMSs predicted by mantle flow models (Fig. 16) (Flament et al., 2022). The statistical significance of the median distance between volcanic eruptions and BMSs (see Fig. 17), expressed as a fraction $f_s$, forms the basis of evaluating the association of plume-related volcanic eruptions and BMSs. $f_s$ close to 1 indicates that the sample distribution is significantly different from the random distribution and that volcanic eruption locations are closer (in a mean sense) to BMS than random points.


For cases OPT2, PMAG and NNR, fa ≈ 0.38, within range for tomographic models (0.32 < fa < 0.52, grey shading in Fig. 18a) while for OPT1, fa ≈ 0.31 is outside the range for tomographic models. This is because the excess density of the basal structure is smaller in OPT1 (+1%) than in other cases (+1.3%), and a greater excess density leads to greater fa (Flament et al., 2022). The match to present-day BMSs as imaged by seismic tomographic models ($\overline{Acc}$) is around 0.71 to 0.72 for the flow models, which is just below the range for tomographic models ($0.74 < (\overline{Acc}) < 0.87$, grey shading in Fig. 18b). ($\overline{Acc}$) is largest for OPT1 (0.72), in which the smaller area of BMSs leads to a larger true negative area, leading to a greater ($\overline{Acc}$). ($\overline{Acc}$) is slightly larger for OPT2 (0.715) than for PMAG (0.71) and NNR (0.706).


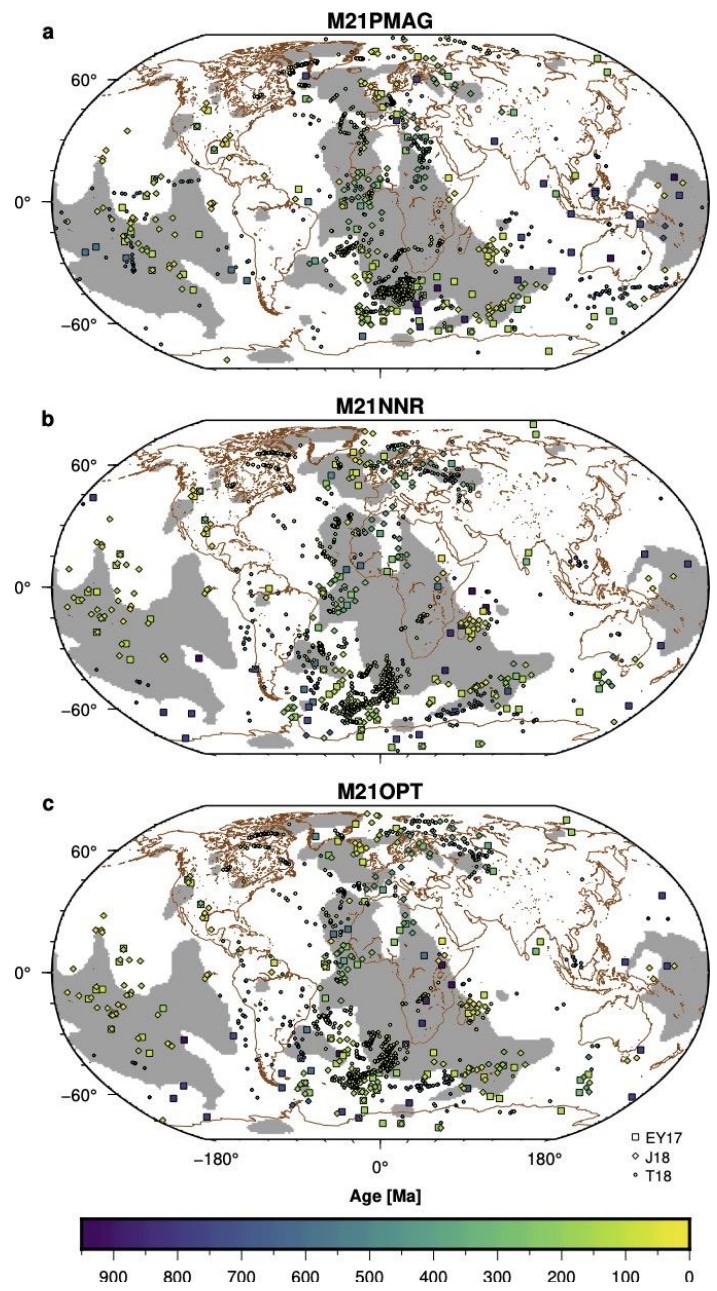

Figure 15. High-velocity (white) and low-velocity (grey) regions revealed by *k-means* cluster analysis between 1000 km and 2800 km depth for seismic tomographic model Savani (Auer et al., 2014), and location of volcanic eruptions (squares, EY17 (Ernst and Youbi, 2017) and diamonds, J18 (Johansson et al., 2018)) and kimberlites (circles, T18 (Tappe et al., 2018)) reconstructed at their time of eruption and shown at present-day using (**a**), tectonic reconstruction PMAG, (**b**),

tectonic reconstruction NNR, and (**c**), tectonic reconstruction OPT. The brown lines are present-day coast lines. Symbols are coloured by age. Robinson projection at Earth's surface.

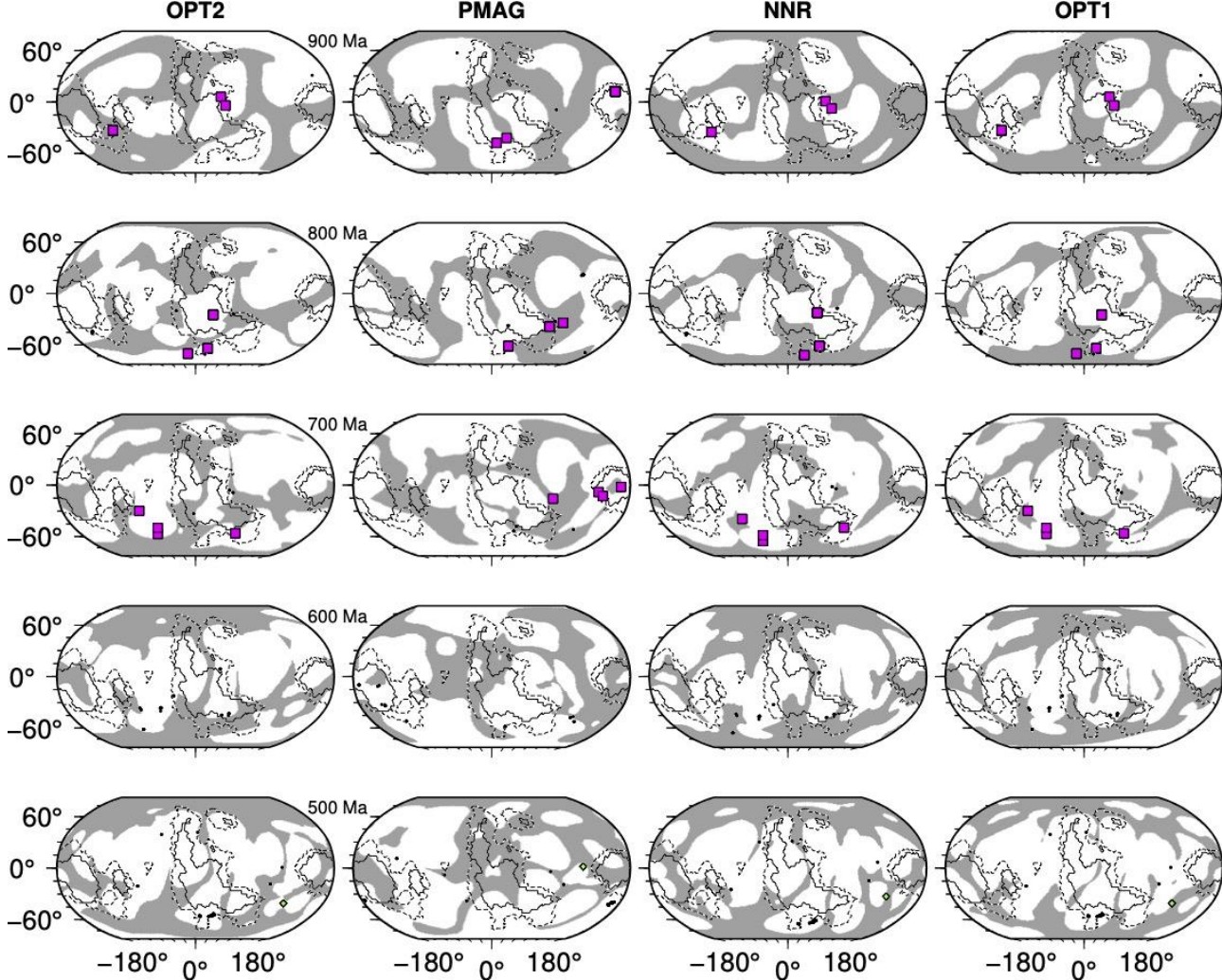


Figure 16a. Low-temperature (white) and high-temperature (grey) regions predicted by plate-mantle models OPT2, PMAG, NNR and OPT1 as indicated, in 100 Myr increments from 900 Ma, and location of volcanic eruptions (magenta squares, EY17 (Ernst and Youbi, 2017) and green diamonds, J18 (Johansson et al., 2018)) and kimberlites (black circles, (Tappe et al., 2018)) reconstructed at the age of interest, shown for eruptions that occurred within 10 Myr of the age of

interest. The black lines indicate a value of five (solid) and a value of one (dotted) in a vote map for low-velocity regions in S-wave tomographic models (Lekic et al., 2012).

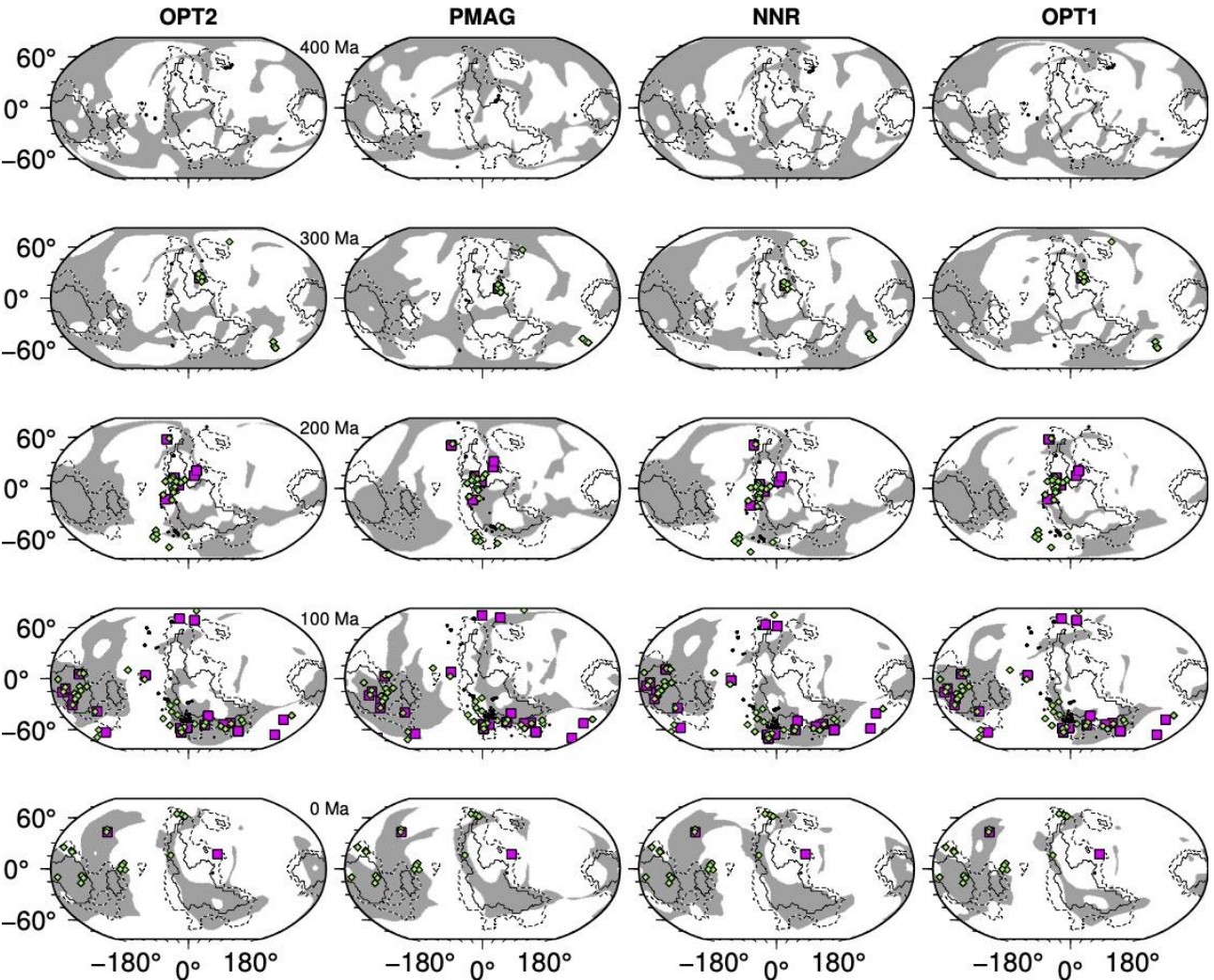

Figure 16b. Figure 16 continued.

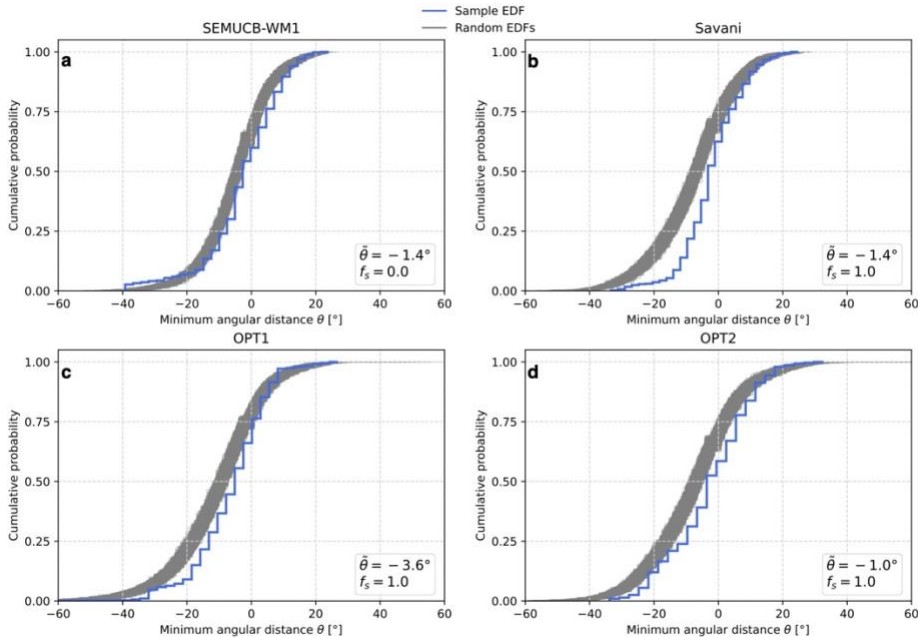


Figure 17. Sample Empirical Distribution Functions (EDFs; blue lines) showing the cumulative probability of minimum

angular distances between 1168 volcanic eruptions and the nearest basal mantle structure over the last 960 Myr for two

tomographic models and two mantle flow models. Grey lines are a series of 1,000 EDFs showing the same quantity for

points randomly distributed around the globe. $\tilde{\theta}$ is the median distance for the sample EDF and fs is the fraction of

random EDFs compared to which the sample EDF passes a statistical test. Results are shown for SEMUCB-WM1 with

reconstruction OPT (a), Savani with reconstruction OPT (b), and mantle flow models OPT1 (c) and OPT2 (d).

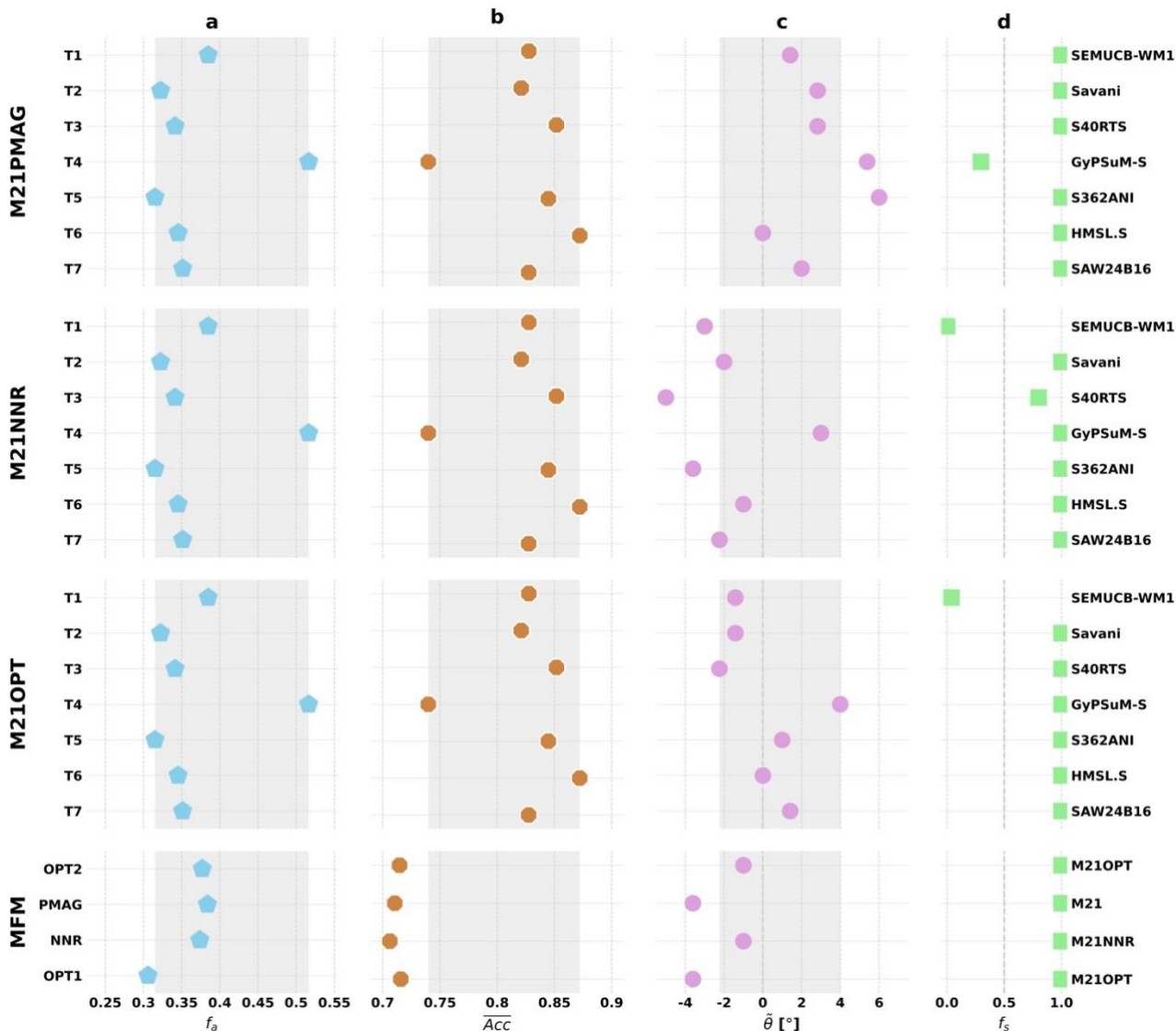

Figure 18. Match between predicted model basal mantle structures, volcanic eruption locations, and tomographic models. **a**, Fractional area $f_a$ of deep mantle cluster maps covered by slow (in tomography) or hot (in flow models, averaged over 960 Myr) basal mantle structures. **b**, Spatial match $\overline{Acc}$ between present-day basal mantle structures for a given case and LLSVPs imaged by tomographic models. **c**, Time-averaged median of minimum angular distances $\tilde{\theta}$ between basal mantle structures and volcanic eruptions from 960 Ma. **d**, Fraction $f_s$ of random EDFs compared to which the sample EDF passes a statistical test (see Fig. 17). In **a-d**, the first three rows show results for a series of tomographic models (stationary LLSVPs) for different reference frames, and the last row shows results for the mantle flow models (mobile

basal mantle structures) considered in this study. "MFM": "mantle flow model"; "T1-T7": tomographic models 1-7. The grey shadings in **a-c** highlight the range of results for tomographic models (based on OPT in c).

We use our optimised absolute plate motion model as a reference for the time-averaged median distance $\tilde{\theta}$ between volcanic products and BMSs ($-2.2° < \tilde{\theta} < 4°$, grey shading in Fig. 18c). Compared to this reference, volcanic products tend to be above BMSs (in a median sense) for reference frame PMAG ($0° < \tilde{\theta} < 6°$), and further outside BMSs for reference frame NNR ($-5° < \tilde{\theta} < 3°$). This primarily reflects that the net rotation for reference frame OPT falls between that for PMAG and NNR (Fig 3). As for mantle flow models, OPT2 and NNR both result in $\tilde{\theta} = -1°$, which is more favourable than PMAG and OPT1 that both result in $\tilde{\theta} = -3.6°$. As for the statistical test, $f_s = 1$ for all four mantle flow models, and for most tomographic models, to the exception of SEMUCB-WM1 for NNR and OPT ($f_s \approx 0$), GyPsuM-S for PMAG ($f_s \approx 0$), and S40RTS for NNR ($f_s \approx 0.75$). This result reflects the large fa for GyPsuM-S ($f_a = 0.52$) and relatively large fa for SEMUCB-WM1 ($f_a = 0.38$). Overall, results for fs confirm that the proximity between volcanic eruptions and BMSs is statistically as consistent when considering mobile BMSs from mantle flow models and when considering stationary BMSs from tomographic models (Flament et al., 2022).

We did not attempt to identify a model that falls within range of tomographic models for ($\overline{Acc}$). While the comparison in cluster space is a useful indication of the match, we did not convert mantle flow models from temperature to density, and we did not apply a tomographic operator (which takes the distribution of earthquakes and seismic stations into account) to the results of mantle flow models. Both these steps would affect the present-day match between mantle flow and tomographic models (Davies et al., 2015a; Schuberth et al., 2009). We note that only one tomographic operator is available for such calculations (Ritsema et al., 2011), and that using this operator on the predicted temperature field has a small effect on ($\overline{Acc}$) (Flament et al., 2017). OPT2 is the preferred model overall in this context, as it fits the location of volcanic eruptions and the present-day structure of the lower mantle better than other models.

### 3.7 Cluster analysis of modelled versus tomographically imaged mantle structure

The spatial match between modelled lower mantle temperature clusters between 1,000 km and 2,800 km depth at present-day from models OPT1 and OPT2 versus seismic tomographic clusters from tomographic models GypsumS, HMSL.S, s40RTS, S3662ANI, SAVANI, saw24b16 and SEMUCB-WM1d (Auer et al., 2014; French and Romanowicz, 2014; Houser et al., 2008; Kustowski et al., 2008; Mégnin and Romanowicz, 2000; Ritsema et al., 2011; Simmons et al., 2010) is illustrated in Figure 13. The relatively larger basal mantle structure buoyancy ratio in model OPT2 relative to our reference model OPT1 results in a larger lateral extent of lower mantle structures (Figs 12a, b). In particular, the anomalously hot lower mantle cluster centred on the Pacific extends significantly beyond the anomalously slow structures captured in the tomography models (Fig. 13b), while the anomalously hot structure underneath Africa is also more extensive in OPT2 than in OPT1. In this instance, the spatial extent of this structure as imaged in the tomography models

is somewhat underestimated in OPT1 (Fig. 13a), highlighting the difficulty of finding a model that matches equally well in all regions.


The model accuracy, i.e. the fraction of correct predictions of all our mantle flow models in terms of the clusters analysed here, is best as compared to tomographic models SAVANI (Auer et al., 2014) and HMSL-S (Houser et al., 2008) at ~75-76%, followed by S40RTS (Ritsema et al., 2011) not far behind at 73-74% (Fig. 14a). GyPSuM-S (Simmons et al., 2010) represents an outlier at the low end with an accuracy of 58-62% because the seismically-slow cluster covers

a larger area in this tomographic model (Fig. 12). GyPSuM-S is different from the other tomographic models used here as it includes constraints from geodynamic and mineral physics (Simmons et al., 2010).

In contrast to the model accuracy, sensitivity is the true positive rate at which the mantle flow model reproduces the geographical distribution of slow clusters in tomographic models, i.e. it represents the percentage of anomalously

slow mantle from tomographic models that is correctly matched by modelled hot mantle temperature anomalies (Fig. 14b). Model sensitivity covers a range of 52-65%, differentiating our preferred model OPT1 clearly from all other models (Fig. 13b). The OPT1 sensitivity is larger than 61% for all tomographic models with the exception of S362ANI (Kustowski et al., 2008) and SAW24B16 (Mégnin and Romanowicz, 2000). In terms of sensitivity, the paleomagnetic and no-net-rotation mantle flow models are the lowest ranked models with a mean of 56-57% averaged across all seven

tomographic models. The top three tomographic models in terms of their match to mantle flow model OPT1 are SEMUCB-WM1 (French and Romanowicz, 2014), SAVANI (Auer et al., 2014) and S40RTS (Ritsema et al., 2011), with a sensitivity between 64 and 66%. In summary, our preferred mantle flow model OPT1 produces the highest mean for accuracy (72%) and sensitivity (61%) averaged across all seven tomographic models while SAVANI (Auer et al., 2014) and S40RTS (Ritsema et al., 2011) consistently produce high scores for models OPT1 and OPT2 for both accuracy and

sensitivity (Figs 14a, b).

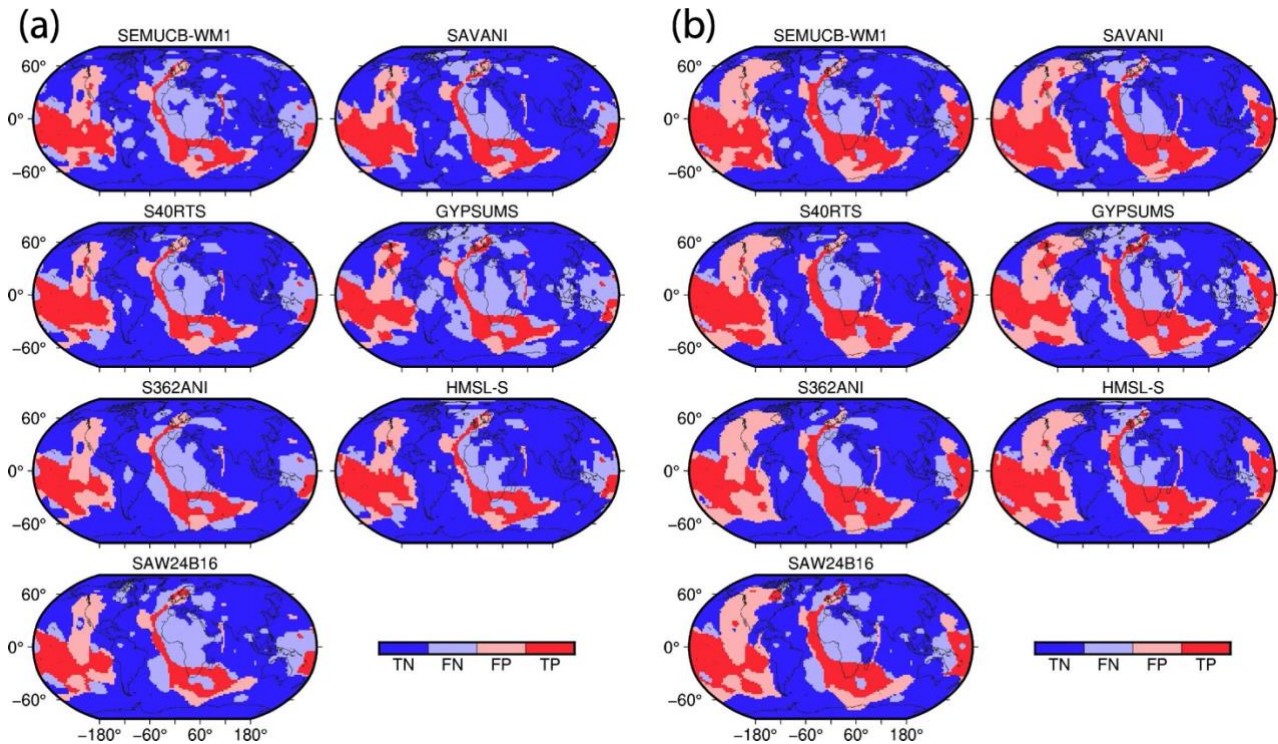


Figure 19. Spatial match between modelled lower mantle temperature clusters between 1,000 km and 2,800 km depth at present-day from models OPT1 (a) and OPT2 (b) versus seismic tomographic clusters from tomographic models SAW24B16, Mégnin and Romanowicz (2000); HMSL-S, Houser et al. (2008); S362ANI, Kustowski et al. (2008); GyPSuM-S, Simmons et al. (2010); S40RTS, Ritsema et al. (2011); SAVANI, Auer et al. (2014); and SEMUCB-WM1, French and Romanowicz (2014). Dark red indicates true positive areas for hot/slow mantle while dark blue indicates true negative regions for cold/fast mantle and grey indicates true negative areas. Light red indicates false positives, i.e. high-temperature mantle clusters paired with low-velocity regions, while light blue indicates false negatives, i.e. low-temperature mantle clusters paired with low-velocity mantle regions. Coastlines are shown in black. TN=True Negative, FN=False Negative, FP=False Positive and TP=True Positive.



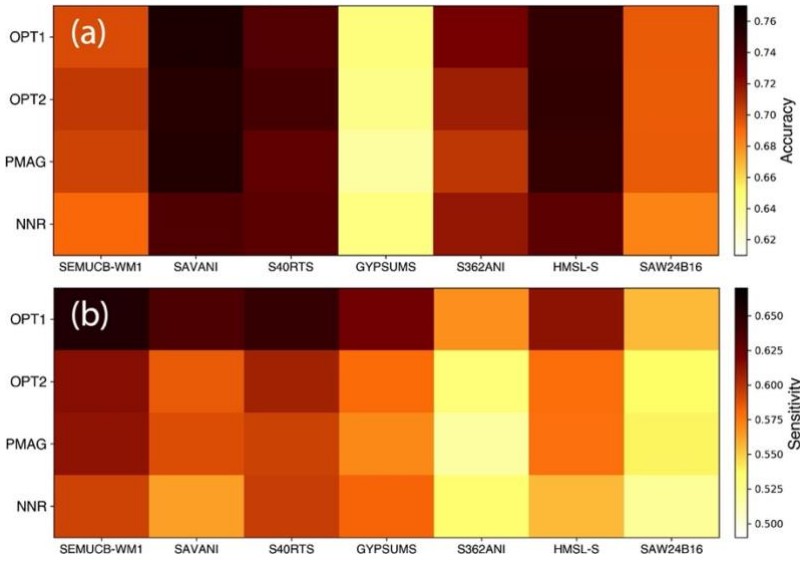

Figure 20. Quantitative match between predicted lower mantle temperature clusters between 1,000 and 2,800 km depth of our four mantle flow models OPT1, OPT2, PMAG and NNR at present-day and seven seismic tomographic clusters from models GypsumS, HMSL.S, s40RTS, S3662ANI, SAVANI, saw24b16 and SEMUCB-WM1 shown as model accuracy (a) and sensitivity (b).

## 4 Discussion

### 4.1 Tectonic rules-based mantle reference frame implications

Our tectonic rules-based absolute plate motion model in a mantle reference frame provides an alternative methodology to constrain both latitudes and longitudes of plates and continents through time. Our optimised model lacks the distinct northward migration of Pangea during and after its assembly featured in the model by Merdith et al. (2021) (compare reconstructions at 400 Ma and 300 Ma on Fig. 2a). This 60° migration, confined mostly to 400-250 Ma, is present in many Pangea reconstructions and was attributed to true polar wander by Le Pichon et al. (2021), an interpretation confirmed by our reconstruction, considering that the migration is absent in our mantle reference frame which does not consider true polar wander. The similarity between our optimised and NNR models confirms that an NNR tectonic reference frame is a reasonable approximation for use in mantle convection models.

In terms of subduction zone migration, our results suggest that the distribution of trench migration, largely confined to a relatively narrow range of ±2 cm/yr during the assembly of Pangea and during its breakup and dispersal (Fig. 5), does not hold for pre-Pangea times, apart from stability during assembly of Rodinia (1000-870 Ma) during the

assembly of Gondwana (650-600 Ma). For other times before the assembly of Pangea, the spread of subduction zone migration rates was significantly larger (Fig. 5), particularly during the period from 600 Ma to 320 Ma (Fig. 4), the "zippy tricentenary", when multiple internal ocean basins evolved, closed and opened rapidly (Fig. 2), coinciding with a peak in passive margin length (Sobolev and Brown, 2019). The beginning of this period follows on closely from the conclusion of the Boring Billion and the end of Neoproterozoic 'snowball' Earth glaciations (Hoffman and Schrag, 2002). The "zippy

tricentenary" might have been initiated by a major burst in surface erosion and subduction zone lubrication event (Sobolev and Brown, 2019), coming to an end when this lubrication became depleted. The "zippy tricentenary" coincides with a global sea level high and a minimum of land area (Kocsis and Scotese, 2021), driven partly by increased oceanic crustal production and an associated increase in the proportion of young, shallow seafloor  (Fig. 8).

After the assembly of Gondwana a number of new ocean basins formed around its periphery, separating it from Laurentia, Baltica, Siberia and other blocks now part of Asia, successively creating and destroying ocean basins including the Iapetus Ocean (Fig. 8). The period was characterised by numerous relatively short subduction zones, which have the capacity to roll back faster than long subduction zones (Schellart et al., 2008).  After 490 Ma the ephemeral Iapetus ocean was replaced by the Rheic Ocean, separating several arc terranes from northern Gondwana, also characterised by fast

trench migration of relatively short subduction zones. Ultimately the difference in the spread of trench migration behaviour (shown as its median absolute deviation in Fig. 5) between the Paleozoic "zippy tricentenary" as compared to the relatively sluggish Late Mesozoic/Cenozoic dispersal of Pangea, reflect that the latter was characterised by a smaller number of relatively long subduction zones which cannot roll back easily, as shown by mantle flow models (Schellart et al., 2007).  The rollback potential of subduction zones wider than 4000 km is limited by the lacking ability of their central

portions to migrate (Schellart et al., 2007).  In contrast much of the Paleozoic Era was characterised by a profusion of relatively short subduction zones in the Merdith et al. (2021) reconstructions, many associated with rapidly evolving internal ocean basins (Fig. 6b).

The most surprising outcome of our mantle reference frame optimisation using a set of tectonic rules is that the

resulting absolute plate rotations produce an orthoversion model in which the centres of Rodinia and Pangea are approximately offset from each other by 90° in longitude (Fig. 2). However, this offset is in opposite direction to that proposed by Mitchell et al. (2012) – they favoured an offset between Rodinia and Pangea to the west to minimise plate velocities, but added that their model does not constrain the sense of motion well.  In our model which minimises plate velocities, Pangea forms about 90° to the east of Rodinia.  The longitudinal sense of motion, constrained by our model,

adds an important dimension to the Rodinia-Pangea orthoversion evolution, as knowing the longitudinal motion of the

plates, and particularly the continents, between successive supercontinents, makes it possible to make geodynamic predictions about plate-mantle interaction, including the effect of dynamic topography and associated relative sea level change affecting the continents involved (e.g. Cao et al. (2019).

In our model Rodinia is located father south (mostly south of 60°S), as compared to the model by Merdith et al. (2021) (Fig. 2). This difference may primarily reflect true polar wander, which our model is agnostic of since it is designed reconstruct plate locations relative to the mantle, without considering whole solid Earth rotations relative to its spin axis. In contrast, the Merdith et al. (2021) model includes motions of plates relative to the mantle as well as true polar wander. Earth's true polar wander history includes a major episode of "Rodinia true polar wander" roughly from 800 Ma to 400

Ma (Mitchell, 2014). This corresponds to the main period during which the latitude and orientation of plates in our model diverges significantly from that of Merdith et al. (2021), providing a plausible explanation for the difference in the reference frames. This inference remains to be tested with independently computed true polar wander models for this time interval.

**4.2 Coupled plate-mantle evolution**

Similarly to previous models, our mantle flow model shows that the geometry and location of basal mantle structures is controlled by subducting slabs (e.g., Bunge et al., 1998; Garnero and Mcnamara, 2008; Zhong and Rudolph, 2015; Bull et al., 2014; King, 2015). Zhong et al. (2007) proposed that the evolution of subduction across supercontinent cycles may cause alternations between degree-1 and degree-2 planform convection. In contrast, Cao et al. (2021a), using

simplified plate motion models to test alternative absolute reference frames since 1 Ga, including an orthoversion and no-net-rotation model, could not find evidence of degree-1 mantle convection forming in their models. Our mantle flow model, which differs from that used in Cao et al. (2021a) in that we use the plate motion model by Merdith et al. (2021) and a novel mantle reference frame, does demonstrate the development of a degree-1 planform, but as a relatively rare occurrence over the last 1 Gyr (Fig. 9a). In our model, a Pacific-centred degree-1 basal structure forms in the leadup to

the assembly of Pangea around 400 Ma (Figs 8, 9), underneath a long-lasting superocean, with the underlying mantle largely protected from descending slabs after ~600 Ma (Figs 9, 10, 12). It takes approximately 200 million years for the basal Pacific mantle region to reflect the absence of subduction above it, with a degree-1 structure initially forming at ~400 Ma. At the same time, an equivalent antipodal structure in the African hemisphere is prevented from forming because it is populated by subduction zones related to the closure of the Iapetus and Rheic oceans, as well as subduction

between Laurussia and Gondwana (Figs 9, 10). Subduction in this region ceases at ~300 Ma, and, mirroring the Paleozoic evolution of the sub-Pacific mantle, it takes nearly 150 million years in our model for a coherent, extensive hot basal structure to form underneath the remnants of Pangea in both models OPT1 and OPT2 (Fig. 9a, b). Large, coherent basal mantle structures thus only form when a lower mantle region is located distal to descending slabs for ~150-200 million years. In general terms, a reconstruction with a reference frame that seeks to stabilise trenches, as is the case in our

optimised model, tends to stabilise mantle structure. Alternative models with a different reference frame (but the same plate boundary configurations) are unlikely to do any better at generating stable deep mantle structures, as can be seen in the more mobile deep mantle structures resulting from the PMAG model (Supp. Animation S6).

The major changes in subduction geometry over the last 1 Gyr therefore dictate that basal mantle structures are 885 ephemeral, and constantly changing in response to subducting slabs pushing against their edges, or flowing over them (Fig. 13). Akin to the extremely heterogenous lower mantle structures seen in the seismic tomography images (e.g., Schuberth et al., 2009; Tkalčić et al., 2015) our model produces bundles of basal mantle upwellings very similar to the thermochemical upwellings enriched in denser than average material interpreted by Davaille and Romanowicz (2020) based on a combination of seismic tomography and fluid mechanic constraints (Fig. 12). In our mantle flow model basal 890 mantle structures are always composed of a network of upwelling structures, from which mantle plumes are emanating. These networks may form and evolve without "LLSVP-like" extensive mantle upwellings, as is the case in our model from 1000-600 Ma, a period without either degree-1 or degree-2 lower mantle structures (Fig. 12, Supp. Animation S11). Their absence during this period reflects the widely distributed, rapidly evolving network of subduction zones, preventing large coherent basal mantle structures to form (Figs 9, 12). Large basal structures form from the coalescence of 895 distributed ridges, as they are being pushed towards each other by descending slabs in nearby regions (Fig. 12, Supp. Animation S11) (see also Bower et al., 2013; Davies et al., 2012).

Most slabs descend into the lower mantle, but we find that slab stagnation occurs both in the transition zone as well as in the mid-lower mantle, around 1000-1400 km depth, as is observed in tomographic models (Shephard et al., 900 2017). However, we point out here that our model does not include phase transitions. The likelihood of stagnation at either depth appears to be increased by fast trench retreat/advance and/or subduction of relatively young lithosphere. Accumulations of slabs below the mid-lower mantle are often detached from shallower slabs and move laterally (Figs 10, 12, Supp. Animation S11). Therefore it cannot be expected that slab accumulations imaged in seismic tomography at depths far below 1400 km can be used as reliable markers for past locations of subduction zones, as implied in the 905 subduction reference frame by Van Der Meer et al. (2010). Our mantle flow model further illustrates the effects of dynamic slab thickening and buckling (Lee and King, 2011) as slabs move from the transition zone into the lower mantle, and slabs also break off (Gerya et al., 2004; Von Blanckenburg and Davies, 1995) either when subduction ceases or due to rapid trench advance or retreat, or to a change in the sign of trench motion (e.g., from advance to retreat). When slabs reach the lowermost mantle, they move laterally towards upwelling regions, pushing hot, low-viscosity basal structures 910 to form steep ridges along their margins (Supp. Animations S3, S4, S11). The coalescence of ridge-like upwellings that are being pushed towards each other by slabs results in enlarged structures with internal and marginal ridges (Fig 12, Supp. Animation S11). Slabs that have sunk to the lowermost mantle gradually heat up and occasionally rise and spread over the edge of basal mantle upwellings and mantle plumes form along basal ridges either in the interior or along the

edges of basal mantle structures (Fig 12, Supp. Animation S11).  The roots of individual plumes are not stationary but migrate in response to the deformation of basal structures, as found previously (e.g., Cao et al., 2021b; Hassan et al., 2016; Arnould et al., 2019). Plume tilt mostly forms in response to relatively fast plate motion and induced sub-horizontal lower mantle flow (Fig 12, Supp. Animation S11).

The spatial match between lower mantle temperature clusters of our preferred model OPT1 with tomographically-imaged lower mantle structure (Fig 13a) demonstrates that our model reproduces the observed large-scale mantle structure quite well. It is noteworthy that the unoptimised model PMAG, not representing a mantle reference frame, reaches an equivalent accuracy to the optimised models OPT1 and OPT2 (Fig. 14a). This reflects that the present-day mantle structure is largely the result of the post-250 Ma subduction history (Flament, 2019) and that the unoptimised versus the optimised models do not show any dramatic differences in the position of plates and subduction zones during this time (compare reconstructions of the two models at 300 Ma and 200 Ma in Fig. 2a). The post-250 Ma differences in the subduction history between these models are not large enough to create any major dissimilarities between the modelled lower mantle structure at present-day. Stark differences between these plate  models are confined to pre-300 Ma times (Fig. 2). The slightly larger excess density of the basal mantle layer in model OPT2 as compared to OPT1 (Table 1) results in a spatially more extensive basal mantle upwellings in OPT2 relative to OPT1 (Figs 9a, b, 13a, b). This slightly improves the match to most seismic tomographic models in the African hemisphere in OPT2, while worsening the match in the Pacific hemisphere (Fig. 13b). In other words, in OPT2, the Pacific LLSVP is somewhat too extensive compared with tomographic images, while in OPT1 the African LLSVP is not sufficiently extensive, as reflected in the variation of the size of the areas labelled as true positive (Figs 13a, b). This may reflect that the excess density of the Pacific versus African LLSVPs is not identical. This is confirmed by a recent analysis of  global shear-wave tomography models, suggesting that the African LLSVP has a relatively lower density and is less stable than its Pacific counterpart (Yuan and Li, 2022).

Upper mantle temperature anomalies through time at ~400 km depth (Fig. 11) illustrate that the largest anomalously hot upper mantle temperatures through time are associated with the formation of the degree-1 convection planform in the Pacific region around 400 Ma, spawning numerous plumes from an extensive network of upwelling basal mantle ridges (Fig. 11, Supp. Animation S11).  However, as all ocean crust formed during this time is now subducted, it is difficult to find observational evidence supporting these model results, even though Doucet et al. (2020a) mapped a pulse of oceanic plume volcanism from 400-200 Ma based on ophiolite data, providing some circumstantial evidence in support of this model result. Cooler than average upper mantle temperatures are associated with continents overriding "slab burial grounds" (Fig. 11).  Mapping such regions in the upper mantle is relevant for understanding magmatism related to upwellings originating from mantle transition zone regions enriched with volatiles from subducting slabs (e.g. Mather et al., 2020; Safonova et al., 2015).

Extensive anomalously cool regions in the upper mantle occur in our model under Siberia, Baltica and North America (420-380 Ma, early to middle Devonian), North America (100-40 Ma, Late Cretaceous to Early Cenozoic) and along eastern/southern Asia and Zealandia (after 100 Ma) (Fig. 11). Devonian magmatism, including intraplate magmatism, in Siberia, Baltica and North America was recently summarised by Ernst et al. (2020). Non-plume related intraplate magmatism across this region may have been driven by subducted volatiles accumulating in the mantle transition zone under these continental regions and driving the formation of volatile-bearing transition zone plumes which have been related to both ocean island basalt-type mafic and felsic melts (Safonova et al., 2015). Post-100-Ma subduction-related intraplate volcanism (including kimberlites) far inland from active trenches has been described both for North America (Currie and Beaumont, 2011; Heaman et al., 2003), eastern Asia (Cao et al., 2021c; Wu et al., 2005) and Zealandia (Mather et al., 2020; Mortimer and Scott, 2020).

The primary differences between our optimised plate-mantle models OPT1 and OPT2 as compared to the PMAG plate-mantle model are driven by the much larger net rotation implicit in the PMAG model and the difference in the reconstructed paleolatitude of Rodinia which is centered on low latitudes in the PMAG model versus a high southern latitude in the optimised plate model (Fig. 2b). This difference illustrates that our model implies a substantial degree of TPW due to the difference between the OPT and PMAG configurations. Using the PMAG plate model as surface condition for a mantle convection model is inherently unreasonable, as the large net rotation embedded in the model, reaching peaks of over 1.2°/Myr, induces significant, lateral displacement of mantle material, which can be readily observed in Supp. Animations S6 and S9. We provide this model merely to demonstrate the difference between mantle and non-mantle plate reference frames on modelled mantle convection. The low-latitude position of Rodinia in the PMAG model prevents the formation of high-latitude LLSVP-like structures, which we observe in models OPT1 and OPT2 from ~600-500 Ma. These generate extensive lower mantle upwellings at high latitudes until the structures are dispersed by migrating subduction zones after 500 Ma and reassemble at low latitudes.

The differences in the modelled history of basal mantle structures, i.e. their location, size and heterogeneity has implications for modelling the Earth's magnetic field through time. LLSVPs increase the insulation of the core-mantle boundary, decrease the temperature gradient and suppress core-mantle boundary heatflow (Li et al., 2018). Glatzmaier et al. (1999) suggested that the polar core-mantle boundary heatflow may be key to driving magnetic reversal frequency. In contrast, Olson et al. (2010) found that the average polarity reversal frequency is sensitive to the total core-mantle boundary heat flow and to the total heat flow at the equator, while reversal frequency may also increase with the amplitude of the boundary heterogeneity. Our basal mantle structure models could be used to evaluate the effect of alternative plate-mantle models on the spatio-temporal patterns of core-mantle boundary heat flow, and magnetic reversal frequency. Such models could also be used to test the validity of alternative reference frames, in terms of how well modelled magnetic reversal frequencies match observed ones.

Further future tests of our absolute reference frames in terms of their suitability as mantle reference frames may include dynamic surface topography models, derived from plate-mantle models. Such models could be compared against geologically mapped continental flooding patterns, following approaches designed to separate effects of eustasy and dynamic topography on continental flooding (e.g., Cao et al., 2019; Müller et al., 2018). In terms of testing alternative orthoversion models, if continents move eastwards after Rodinia breakup, as in our optimised mantle reference frame, one would expect the eastern portions of continents to be flooded first as they move towards dynamic topography lows associated with subduction zones to the east of Rodinia. In contrast, if continents move westwards after Rodinia breakup as suggested by Mitchell et al. (2012) and implemented as a model with evolving plate boundaries by Cao et al. (2021a), one would expect to see the western portions of continents to be inundated first after Rodinia breakup.

## 5 Conclusions

We have used "tectonic rules" to optimise absolute mantle reference frame devoid of unreasonably large lithospheric net rotation, excessive subduction zone migration rates, and excessive speeds for plates hosting large continents. Our model results in net rotation consistently below 0.25°/Myr, while trench migration scatter is substantially reduced compared with the unoptimised model. Trench motion scatter is confined to a relatively narrow range during Pangea stability and dispersal, mostly between -1 cm/yr (trench advance) and 2 cm/yr (trench retreat). In contrast, the period between 600 Ma and 320 Ma stands out as the most dynamic time in terms of ocean basin evolution and subduction zone migration in the last billion years, with relatively short, highly mobile subduction zones dominating – we propose to call this period the "zippy tricentenary". Our model independently confirms an orthoversion evolution from Rodinia to Pangea as proposed by Mitchell et al. (2012), but involving an eastward shift of their respective centres, not westward as previously suggested. We show that the emplacement history of large igneous provinces and kimberlites is consistent with deep mantle plumes associated with basal mantle structures, thus illustrating that their eruption history does not demand stationary LLSVPs, as previously hypothesised (Torsvik et al., 2012).

Our mantle flow model is driven by the imposed plate motions and subduction history and results in a succession of deep mantle states without or with large basal mantle structures akin to present-day LLSVPs. Our numerically modelled basal mantle structures bear a striking resemblance to the mantle tomographic images by Davaille and Romanowicz (2020). Their tomographic model, together with laboratory experiments, led to the view that LLSVPs are composed of bundles of thermochemical upwellings, whose shape is controlled by subduction history, resulting in a position and geometry of LLSVPs that are time dependent (Davaille and Romanowicz, 2020). This view is supported by our work, which explicitly links the evolution of the plates and plate boundaries over time with mantle structure evolution. Our model records five distinct intervals of mantle convection evolution over the last 1000 Myr. Initially, a broad network

of basal ridges forms between 1000 Ma and 600 Ma, followed by the formation of a short-lived degree-2 basal mantle structure centred on the north and south pole between 600 Ma and 500 Ma. It is superseded by a transitional phase during which the north polar basal structure migrates southward and gradually morphs into an extensive Pacific-centred basal structure while the south polar structure is dissected by subducting slabs and disintegrates into a network of ridges between 500-400 Ma. Subsequently a Pacific-centred degree-1 structure forms and is stable between 400 Ma and 200 Ma, which is superseded by a basal degree-2 mantle structure after ~160 Ma. This succession of mantle states is distinct from previously proposed models. Our Solid Earth Evolution Model for the last 1000 million years (SEEM1000) that can be analysed, tested and modified to provide insights into the history of magmatism, the history of the Earth's geodynamo and magnetic field reversals, mineral resources, sea level change and biological evolution, forming the foundation for a multitude of spatio-temporal data analysis approaches.

**Code Availability**

The codes used for this paper are available on the public EarthByte and GPlates github sites. This includes absolute plate motion model optimisation code at https://github.com/EarthByte/optAPM, CitcomS software for mantle convection modelling at https://github.com/EarthByte/citcoms, and GPlates/pyGPlates at https://github.com/gplates for viewing and manipulating plate reconstructions.

**Data availability**

The optimised plate model, including paleo-oceanic crustal age grids, used in this paper is available on the EarthByte webdav site at https://www.earthbyte.org/webdav/ftp/Data_Collections/Muller_etal_2022_SE/. Mantle flow model configuration files and outputs are available on zenodo (see below).

**Video supplement**

Supplementary videos are available on zenodo (see below).

**Supplement link**

The data and video supplements are available on zenodo at 10.5281/zenodo.5801084.

**Author contribution**

RDM conceived and coordinated the research underpinning the paper, and wrote the manuscript. JC contributed to the design of the absolute plate model optimisation code, executed the optimisation runs and contributed to the manuscript. MT wrote the original optimisation code and contributed to the manuscript. SEW contributed to the plate model optimisation code, wrote Jupyter notebooks to visualise results, and contributed to manuscript. XC executed the mantle flow models, contributed to the analysis of the results and to the manuscript. NF oversaw the design of the mantle flow

models, contributed to the model analysis and to the manuscript. OFB visualised the mantle flow models, SZ and AM contributed to the plate model and the manuscript.

**Competing interests**

The authors declare that they have no conflict of interest.

**Acknowledgments**

This research was supported by Australian Research Council grants LP210100173, LP170100863 and DE210100084 and supported by the Australian Government's National Collaborative Research Infrastructure Strategy (NCRIS), with access to computational resources provided by the National Computational Infrastructure (NCI) through the National Computational Merit Allocation Scheme and through the Sydney Informatics Hub HPC Allocation Scheme, which is supported by the Deputy Vice-Chancellor (Research), University of Sydney and ARC grant LE190100021.

Table 1. Parameters for mantle flow models.

| Parameters | Value |
|---|---|
| Rayleigh number | 7.8e7 |
| Dissipation number | 1.56 |
| Earth radius | 6371 km |
| Mantle thickness | 2867 km |
| Initial slab depth | 1000 km |
| Basal layer thickness | 113 km |
| Coefficient of thermal expansion | $3 \times 10^{-5}$ K$^{-1}$ |
| Reference density | 4000 kg m$^{-3}$ |
| Acceleration of gravity | 9.81 m s$^{-2}$ |
| Temperature difference between surface and CMB | 3100 K |
| Reference viscosity | 1.1e21 Pa s |
| Thermal diffusivity | $1 \times 10^{-6}$ m$^2$ s$^{-1}$ |
| Depth-dependent viscosity pre-factor | 0.02,0.002,0.02,0.2 (above 160 km, 160-310 km, 310-660 km and below 660 km) |

| Compositional viscosity pre-factor | 1, 100, 10 for ambient mantle, continental lithosphere, and basal layer. |
|---|---|
| Activation energy | 283.5 kJ mol$^{-1}$ |
| Activation volume | 2.1 cm$^3$ mol$^{-1}$ |
| Temperature offset | 496 K |
| Heat capacity | 1200 J kg$^{-1}$ K$^{-1}$ |
| Internal heating rate | 33.6 TW |
| Model warmup time | 250 Myr |
| Buoyancy ratio for basal layer | OPT1: 0.25 (1% excess density), OPT2, NNR, PMAG: 0.325 (1.3% excess density) |
| Hot basal structure (Paraview visualization) | Mantle hotter than layer average by non-dimensional value 0.1 (310 K) |
| Slab (Paraview visualization) | Mantle colder than layer average by non-dimensional value 0.05 (155 K) |

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
