# Peer review of "A tectonic-rules based mantle reference frame since 1 billion years ago – implications for supercontinent cycles and plate-mantle system evolution"

_Solid Earth, 2021_

## Referee Comment (RC2)

**Title:** A tectonic-rules based mantle reference frame since 1 billion years ago – implications for supercontinent cycles and plate-mantle system evolution.
**Journal:** Solid Earth
**Authors:** R. D. Müller et al.
**Reviewer:** Rhodri Davies (ANU).

In this manuscript, authors: (i) present a new global plate tectonic reconstruction from 1 Ga to the present day, in the mantle reference frame, that was developed using tectonic rules-based optimisation strategy; (ii) use this as a constraint for mantle flow models, to understand the evolution of mantle structure over this time, thus connecting tectonic motions at Earth's surface to the underlying mantle. These topics will obviously be of interest to the readership of Solid Earth.

There are several important results in the paper, three of which seem particularly far-reaching:

1. *Supercontinent cycles:* results support an orthoversion evolution from Rodinia to Pangea, with Pangea offset approximately 90° eastwards relative to Rodinia. This is very important result: it is the opposite sense of motion compared to previous studies based solely upon paleomagnetic data and is, obviously, more consistent with the "tectonic-rules" incorporated through the optimisation.

2. *The evolution of mantle structure in response to these extended plate motion reconstructions:* model predictions point towards 5 main stages for deep mantle structure over the past 1 Byr: (i) a broad network of hot basal ridges between 1000 and 600 Ma; (ii) the formation of a short-lived degree-2 basal mantle structure with upwellings centred on the poles from 600-500 Ma; (iii) a transitional phase during which the north polar basal structure migrates southwards and gradually morphs into an extensive Pacific centred basal structure, while the south polar structure is dissected by subducting slabs and disintegrates into a network of ridges between 500-400 Ma; (iv) a Pacific-centred degree 1 structure that is stable from 400 – 200 Ma; and (v) a basal degree 2 structure, post 160 Ma, with upwellings centred beneath the Pacific and African domains. This succession of mantle states is distinct from previously proposed models, as would be expected given the differences in the plate motion histories. It implies a mobile deep mantle and has important implications for our understanding of Earth's thermo-chemical evolution and how this links to the surface.

3. *Reference frames:* The NNR rotation reference frame, enforced by most existing *mantle circulation models*, is reasonably ok, which adds confidence to the results of these previous studies.  This is a relief!

Given the above, I feel that the manuscript is clearly worthy of publication in Solid Earth. However, there are a few changes that I would recommend, which are outlined below, that I feel would further strengthen the manuscript. My main comments are presented first, followed by more minor suggestions.

Thank you for the opportunity to review this work. I hope the authors find my comments useful.

Best wishes,

Rhodri

**Main comments:**

1. As noted above, one of the main results from the optimised plate tectonic reconstructions is the orthoversion evolution from Rodinia to Pangea, albeit in a different direction to earlier work by Mitchell *et al.* (2012). This is potentially highly significant and, if correct, it is of fundamental importance for our understanding of coupled plate-mantle models. Given its importance therefore, I would urge the authors to make more of an effort to find independent observations that support their findings. Is this result supported by other observations? If such observations are currently lacking, or the authors are unaware of any, at the very least they should present testable hypotheses from their models that would allow others to later test the validity of the different results. What is it about the coupled plate-mantle models that differ for example, between the OPT end member and the PMAG end member? Would the models, for example, predict a different dynamic topography evolution? Would your models predict a different magmatic record in the continents (where a record of such magmatism could be preserved)? What else is different? I think it's vital to draw out these differences so that others can build on your work.

2. Authors spend a lot of time comparing their model predictions to present-day mantle structure, as imaged through seismic tomography. I will outline my concerns with the specifics of approach that has been used below. However, in the context of this paper, such comparisons are almost irrelevant: I found them to be a distraction from the paper's main message. Previous studies have shown that deep mantle structure is only sensitive to the past 200-250 Myr of plate motion histories – this has been pretty much clear since the work of McNamara & Zhong (Nature, 2005) and perhaps even Bunge et al. (PTRS, 2002). Given that the different reconstructions examined here are all similar at 200 Myr, there really is not much to be gained (at least in my opinion!) by examining present-day structure. The focus of results and discussion should be more on the differences predicted between each scenario as a function of time, as suggested in point 1 above. The authors almost acknowledge this themselves when stating – "it is noteworthy that the unoptimized model PMAG, not representing a mantle reference frame, reaches an equivalent accuracy to the optimised models OPT1 and OPT2. This reflects that the present-day mantle structure is largely the results of the post-250 Ma subduction history." I'd suggest removing these comparisons or transferring across to the supplement.

3. Tectonic rules: I realise these have been outlined by Tetley et al. (2019), but it would be good to see a little more of a summary in the current paper, to provide valuable background material. In the context of the current paper, it would also be good to comment on whether such assumptions/rules are reasonable, back to 1 Gyr.

**Minor points:**

1. Abstract – the approach of Bower et al. (2015) amounts to more than a surface boundary condition (thermal structure, dip angle etc… are also imposed). I would therefore recommend changing line 24 to reflect this. Perhaps "use it as a constraint on mantle flow models", or similar?

2. Line 65 – … a combination of relative plate motion and *constraints provided by* mantle convection models.

3. Line 80-85 it is stated that the assumption of fixed LLSVPs is based on an apparent correlation between the reconstructed eruption sites of LIPs and kimberlites, from the work of Torsvik, Burke and others. However, using powerful statistical approaches, a number of studies (e.g. Austermann et al. GJI, 2014; Davies et al. EPSL, 2015), have shown that this correlation is not robust, whilst a follow up study by Doubrovine et al. (2016) essentially shows the same (i.e. you cannot conclusively state that plumes are forming at edges over LLSVP interiors). I find it surprising that these studies, which support the authors conclusions for mobile deep mantle structure, are not cited or discussed. Alongside the models presented, these studies provide a solid basis for challenging the fixed LLSVP hypothesis of Torsvik and others.

4. Line 91 – probably fair to cite work by Bull et al. (EPSL, 2009) and Davies et al. (EPSL, 2012) here too.

5. I found the jump in logic from Line 101 to Line 102 hard to follow at first. Do "alternative modes of supercontinent formation" really belong in a subsection on LLSVPs? Having read it a few times, I see

the link, but perhaps a separate section, or a sentence explicitly connecting these two aspects, would be helpful.

6. Line 232 – model setup – you limit the age of the lithosphere to 80 Myr when constructing the thermal structure of plate, but still use a half-space model. Why is this? Why not use a plate model, where thickness changes are small beyond this age anyway? Are your results sensitive to this age? If so, it's probably worth explicitly acknowledging that this is the case.

7. Line 236 – just a flag that the CMB temperature used in these models is very much towards the lower end of current estimates. The Di is also higher than I'd have expected. I'd recommend that authors provide a justification for their choices.

8. Line 247 – 256 – there seems to be spurious use of bold font in places.

9. Viscosity: I find it difficult to convert that beautiful (!) equation describing your viscosity into an understanding of the range of values in the model and their depth and lateral sensitivity. Could you add a plot showing the depth average and range of values? This will help a reader to place your results in the context of other studies with different rheological approximations.

10. Comparisons between model predictions and imaged structure: as noted in the main points above, I do not feel that these comparisons add anything to this paper and find them a little distracting. Dropping these comparisons would free up space to discuss your exciting results in more depth. As well as this, I have a major concern with how such comparisons are undertaken. Seismic velocity is non-linearly dependent on temperature, composition and phase. Furthermore, tomographic models have limited and uneven resolution. None of these important factors are considered in the comparisons that you present. Several previous studies (e.g., Bull et al. EPSL 2009; Schuberth et al. G3, 2009; Davies et al. EPSL, 2012) demonstrate that they need to be considered when comparing models with tomography.

11. Just a comment. I REALLY liked Figures 2 and 3. They were very useful for a geodynamicist that is not an expert in plate motion reconstructions. It was valuable to be able to directly compare the different reference frames.

12. Lines 394 – 410. This is a very interesting insight. I don't know the answer of the top of my head and haven't had time to appraise the literature, but are these trends supported by models that examine the evolution of trench retreat under various scenarios (I'm thinking of work by Goes, Garel, Van Hunen, Capitanio, Moresi, Holt, for example)?

13. Line 407 – who doesn't get excited by the "zippy tricentenary"? It's not a term I'll forget in a hurry!

14. Figure 4 – remove duplicate scale bars (unless I'm missing something)?

15. Line 443 – you mention that there are some periods of relatively large RMS speeds and attribute these to potential artefacts in the reconstruction. Could you say a little more here to help a non-expert? What type of artefacts are these? And why are you confident that they are not present at other times?

16. Line 470 – you use the term ridges and nodes. What is meant by nodes here? Are they simply ridge intersections? If so, perhaps use ridge intersections instead, or define nodes on first use (given that you also use nodes in a different context elsewhere in manuscript).

17. Line 490: … basal mantle structure *with upwellings* centered on the north and south pole…

18. Fig. 11 – I could not easily make out the bright red dots. Perhaps enlarge? Or add crosses or similar?

19. Line 643 – it is explicitly mentioned here, but it is also mentioned elsewhere in the paper: short subduction zones have the capacity to roll back faster than long subduction zones. In general, this is true, but I think the reality is a little more nuanced. If the downgoing plates are young, trench retreat is limited, even for short subduction zones. In other words, the magnitude of trench retreat does not only depend on the length of a subduction zone, but also its age (as well as complications arising from overriding plates etc…). This is explicitly covered in a pre-print here https://www.essoar.org/doi/10.1002/essoar.10508606.1. Potentially something that's worth looking at further down the line in your extended reconstructions is whether you see evidence for these dependencies in your reconstructions.

20. Line 696 – strcturee – structure.

21. Line 710-712 – it's probably fair to cite work by Davies et al. (EPSL, 2012) and Bower et al. (G3, 2013) here.
22. Lines 714-720 – with the comparisons of slab depths it's important to acknowledge that your models do not include phase transitions, which are important in dictating the form of slab transition-zone interaction.
23. Line 789 – I think a little too much credit is given to the study of Davaille and Romanowicz (2020) here. I'm not denying it's a wonderful study, but it builds on concepts and inferences from many previous studies that are not cited. I would recommend perhaps giving some credit to some earlier work in this area, alongside the work of Davaille and Romanowicz.

Anyway, that's it from me! Hopefully these points are useful and will allow the authors to further improve what I felt was already an excellent study.

---

## Author Response (AR2)

Response to reviewers' comments (replies in italics)

**Review 1 (Scott King)**

First off, this is a long paper with many detailed components and one could easily go down a rabbit hole picking at various assumptions one doesn't like. In spite of this there are several fairly straightforward conclusions. The authors frame this work as applying a "tectonic rules" based approach however I think a more useful description would be to say that they are testing a set of assumptions. Because they compare their approach with a series of other approaches, it strikes me that this is a test. The tectonic rules are (lines 198-201): "(1) rates of net lithospheric rotation (NR) are minimized but non-zero, (2) global trench migration velocities are minimized, favouring trench retreat over trench advance, (3) spatio-temporal misfit between plate motion model and present-day hotspot chains is minimized, and (4) global continental median plate speed remains < 6 cm/yr, based on continental plate speed statistics reported in Zahirovic et al. (2015)." I wonder, recognizing that these "rules" are really "assumptions" and/or general results of geodynamic models that may or may not be unique, whether all of these rules are necessary or whether one or two of them are sufficient to produce the results reported in the work. This is important because these "rules" are not really hard and fast rules.

*We agree that our "rules" could be regarded as "assumptions". However, rules are nothing but principles that have been established to govern some process or activity. By calling such principles rules, it does not necessarily follow that everyone agrees that these rules should be followed or that they are universally applicable. They are the rules that govern the optimisation procedure we have developed, because we believe that they are important for how plate tectonics works based on cumulative evidence from published data and geodynamic models. Of course not everyone may agree, but we prefer to retain the use of the expression "rules" in our paper.*

First (lines 777-779), these are the assumptions the authors used to constrain the model, that the model results are consistent with the assumptions is useful (e.g., you didn't screw up the optimization) but they are not really conclusions.

*We have removed this sentence from the conclusions.*

Second, the authors focus on the LLSVPs in the conclusions, which are problematic for two reasons:

As the authors state (lines 736-742) "It is noteworthy that the unoptimised model PMAG, not representing a mantle reference frame, reaches an equivalent accuracy to the optimised models OPT1 and OPT2 (Fig. 14a). This reflects that the present-day mantle structure is largely the result of the post-250 Ma subduction history (Flament, 2019) and that the unoptimised versus the optimised models do not show any dramatic differences in the position of plates and subduction zones during 740 this time (compare reconstructions of the two models at 300 Ma and 200 Ma in Fig. 2a). The post-250 Ma differences in the subduction history between these models are not large enough to create any major dissimilarities between the modelled lower mantle structure at present-day." This is one of the most important aspects of this work and it shows that the "rules" approach used here isn't really significant to achieve the LLSVP structure.

*We agree with this point, and have now added an alternative method of assessing the success of the optimised model, following the method used in a recently published paper by Flament et al. (2022). This is a statistical method used to assess how consistent the history of LIP volcanism through time is with modelled mobile basal mantle structures, as compared to fixed ones, over the last billion years. Also, to further clarify, the mantle reference frame rules were not primarily introduced to be able to reproduce the current LLSVP structures. They are clearly not necessary for that purpose. They were instead introduced because reference frames that do not satisfy our optimisation criteria, especially in terms of net rotation and subduction zone migration, introduce many obvious artefacts in plate-mantle models, including unreasonable lateral displacement of mantle material (in response to unreasonable net rotation) and unreasonable behaviour of mantle flow associated with subduction zones that present unreasonable lateral motions.*

In the geodynamic modeling, the imposed plate and subduction motions/locations, dominate the flow and significantly disrupt the balance of forces. The fact that the authors don't see a difference between the two different density scaling factors for the LLSVPs is consistent with this. The statement, "demonstrating that the excess density of the basal mantle layer plays a secondary role, in comparison to the imposed plate motion history" may or may not be true. It is an artifact of the modelling approach, not a conclusion. That LLSVPs get moved around by the plate slab system isn't really novel. It was described in Bull et al. (2014) and in King (2015) as well as by many other authors, some of whom are cited at lines 677-679. In fact, in King (2015) I don't impose plate velocities so this is somewhat unique, although to be transparent, many people don't like the approach I use to create long-wavelength plate-like flow.

*We agree and have added the references by Bull et al. (2014) and in King (2015). We agree that there are other papers also concluding that subduction motions/locations dominate mantle flow, but it would be unreasonable to expect having all these papers cited. Our list of cited papers is preceded by "e.g.", implying that this is just a representative subset of papers. We have modified the statement "demonstrating that the excess density of the basal mantle layer plays a secondary role, in comparison to the imposed plate motion history" to "suggesting that the excess density of the basal mantle layer may play a secondary role, in comparison to the imposed plate motion history", following the reviewer's suggestion that this may or may not be true.*

Next, one of the more interesting conclusions is buried in the middle of the paper (lines 319-322) and should be repeated in the conclusions: "This comparison provides an important insight, namely that the simple lithospheric no-net-rotation rule used to produce the NNR model produces results that are not dramatically different from a model optimised by a set of more general tectonic rules." The next sentence (lines 322-324) speaks to the importance of the result, "This is important because NNR models have been frequently used in tectonic and mantle flow models for practical reasons (e.g., Mao and Zhong, 2021; Zhong and Rudolph, 2015; Behn et al., 2004; Kreemer and Holt, 2001) in the absence of other available mantle reference frames." This gets to the heart of my concern in the first paragraph, which rules are critical and which rules are not? Do we need them all? Does imposing one enforce the others?

*We have now added this explanation: In terms of the relative importance of plate motion optimisation parameters, our results suggest that minimising net rotation is the most important one, with minimising subduction zone migration of secondary importance, as minimising net rotation also reduces subduction zone migration to some extent (also see Müller et al. (2019) for a*

*discussion of the effect of changing the relative weight of these parameters). Preventing the speed of continents to exceed continental speed limits is the least important parameter. We introduced it to ensure that large swathes of synthetically reconstructed ocean floor would not result in a minimal net rotation solution that imposes unreasonable motions on the smaller continental regions.*

Finally, the interesting aspects of the reconstruction (lines 793-804) is the second part of the paragraph starting with LLSVPs and the present day pattern (which as I mention above) the authors point out is really controlled by the past 250 Myrs and not the period covered by the tectonic assumptions. The reconstruction differs from the assumption from the Oslo researchers that the piles are more or less stationary and from the results of Zhong, Rudolph, Zhang (and others in the Colorado group) that should an oscillation between degree 1 and degree 2 anomalies but (as I recall) these were mostly in the equatorial region. This work does see some transition between degree 2 and degree 1 (lines 705-710) and the evolution appears to be quite different than what has been seen in previous work. I'm something of a visual person and it would really be helpful to find a sketch diagram of the LLSVP locations through time. I realize there are different figures with temperatures in the deep mantle (Figure 9, 11, 12, 13) but it would be helpful to link the text more closely to the figures. The upper mantle structure (Figure 10) feels like a distraction and not really part of the story. It certainly doesn't help me. With all those figures I'm not sure they add to my understanding or overwhelm. I really think this part of the paper could be streamlined.

*We agree, and have now added a new figure that is essentially a set of "sketches" that show the distribution of the African and Pacific LLSVPs through time, as well as LLSVP material that is not part of either the African or Pacific LLSVP. (Fig. 12). Further back in time (early Paleozoic and Proterozoic) the geometry of the LLSVPs becomes quite complex, such that the relative proportion of "non-African" and "non-Pacific" LLSVP material increases. The sketches should help with visualising LLSVP geometries through time.*

*In terms of upper mantle structure, we prefer to retain these figures in the paper, as there is a large community seeking to connect the evolving upper mantle thermal structure to intraplate magmatism. Therefore, these figures should be of interest to a range of readers in the field of geology.*

The idea of two thermochemical piles at the poles is quite interesting and could/should have interesting implications for the dynamo. There is no mention of that here.

*We agree and have added a paragraph on this topic as follows:*

*The differences in the modelled history of basal mantle structures, i.e. their location, size and heterogeneity has implications for modelling the Earth's magnetic field through time. LLSVPs increase the insulation of the core-mantle boundary, decrease the temperature gradient and suppress core-mantle boundary heat flow (Li et al., 2018). Glatzmaier et al. (1999) suggested that the polar core-mantle boundary heat flow may be key to driving magnetic reversal frequency. In contrast, Olson et al. (2010) found that the average polarity reversal frequency is sensitive to the total core-mantle boundary heat flow and to the total heat flow at the equator, while reversal frequency may also increase with the amplitude of the boundary heterogeneity. Our basal mantle*

*structure models could be used to evaluate the effect of alternative plate-mantle models on the spatio-temporal patterns of core-mantle boundary heat flow, and magnetic reversal frequency. Such models could also be used to test the validity of alternative reference frames, in terms of how well modelled magnetic reversal frequencies match observed ones.*

When I look at Table 1, I wonder why there is a thermal diffusivity but no coefficient of thermal expansion, etc. It seems somewhat odd to call out this one material property and not list them all.

*We had refrained from listing all parameters involved, as they have been listed in previous papers that this paper builds upon, especially (Flament, 2019). However, we have now followed the reviewer's suggestion and listed all relevant parameters in Table 1.*

**Review 2 (Rhodri Davies)**

In this manuscript, authors: (i) present a new global plate tectonic reconstruction from 1 Ga to the present day, in the mantle reference frame, that was developed using tectonic rules-based optimisation strategy; (ii) use this as a constraint for mantle flow models, to understand the evolution of mantle structure over this time, thus connecting tectonic motions at Earth's surface to the underlying mantle. These topics will obviously be of interest to the readership of Solid Earth.

There are several important results in the paper, three of which seem particularly far-reaching: *Supercontinent cycles:* results support an orthoversion evolution from Rodinia to Pangea, with Pangea offset approximately 90° eastwards relative to Rodinia. This is very important result: it is the opposite sense of motion compared to previous studies based solely upon paleomagnetic data and is, obviously, more consistent with the "tectonic-rules" incorporated through the optimisation.

*The evolution of mantle structure in response to these extended plate motion reconstructions:* model predictions point towards 5 main stages for deep mantle structure over the past 1 Byr: (i) a broad network of hot basal ridges between 1000 and 600 Ma; (ii) the formation of a short-lived degree-2 basal mantle structure with upwellings centred on the poles from 600-500 Ma; (iii) a transitional phase during which the north polar basal structure migrates southwards and gradually morphs into an extensive Pacific centred basal structure, while the south polar structure is dissected by subducting slabs and disintegrates into a network of ridges between 500-400 Ma; (iv) a Pacific- centred degree 1 structure that is stable from 400 – 200 Ma; and (v) a basal degree 2 structure, post 160 Ma, with upwellings centred beneath the Pacific and African domains. This succession of mantle states is distinct from previously proposed models, as would be expected given the differences in the plate motion histories. It implies a mobile deep mantle and has important implications for our understanding of Earth's thermo-chemical evolution and how this links to the surface.

*Reference frames:* The NNR rotation reference frame, enforced by most existing *mantle circulation models*, is reasonably ok, which adds confidence to the results of these previous studies. This is a relief!

Given the above, I feel that the manuscript is clearly worthy of publication in Solid Earth. However, there are a few changes that I would recommend, which are outlined below, that I feel would further strengthen the manuscript. My main comments are presented first, followed by more minor suggestions.

Thank you for the opportunity to review this work. I hope the authors find my comments useful. Best wishes,
Rhodri.

**Main comments:**
As noted above, one of the main results from the optimised plate tectonic reconstructions is the orthoversion evolution from Rodinia to Pangea, albeit in a different direction to earlier work by Mitchell *et al.* (2012). This is potentially highly significant and, if correct, it is of fundamental importance for our understanding of coupled plate-mantle models. Given its importance therefore, I would urge the authors to make more of an effort to find independent observations that support their findings. Is this result supported by other observations? If such observations are currently lacking, or the authors are unaware of any, at the very least they should present testable hypotheses from their models that would allow others to later test the validity of the different results. What is it about the coupled plate-mantle models that differ for example, between the OPT end member and the PMAG end member? Would the models, for example, predict a different dynamic topography evolution? Would your models predict a different magmatic record in the continents (where a record of such magmatism could be preserved)? What else is different? I think it's vital to draw out these differences so that others can build on your work.

*There are currently no observations that can be used in a straightforward manner to distinguish between alternative orthoversion scenarios. If one wanted to impose an orthoversion scenario as that proposed by Mitchell et al. (2012) on our plate model, one would, of course, introduce an enormous rate of net rotation, which would make this model intrinsically invalid as a mantle reference frame. One would need to design a completely different plate model to be able to design a version of the Mitchell et al. (2012) model with minimal net rotation. This might be possible but is outside of the scope of our current work. However, we have now added three paragraphs addressing these questions how the models differ, and how they could be further tested, including the use of dynamic topography – see below. Comparing modelled dynamic topography to observations back to the Neoproterozoic would be a huge, multiyear research project by itself.*

*The primary differences between our optimised plate-mantle models OPT1 and OPT2 as compared to the PMAG plate-mantle model are driven by the much larger net rotation implicit in the PMAG model and the difference in the reconstructed paleolatitude of Rodinia which is centered on low latitudes in the PMAG model versus a high southern latitude in the optimised plate model (Fig. 2b), primarily reflecting the lack of True Polar Wander in our mantle reference frame. Using the PMAG plate model as surface condition for a mantle convection model is inherently unreasonable, as the large net rotation embedded in the model, reaching peaks of over 1.2° per m.y., induces significant, unreasonable lateral displacement of mantle material, which can be readily observed in Supp. Animations S6 and S9. We provide this model merely to demonstrate the difference between mantle and non-mantle plate reference frames on modelled mantle convection. The low-latitude position of Rodinia in the PMAG model prevents the formation of high-latitude LLSVP-like structures, which we observe in models OPT1 and OPT2 from ~600-500 Ma. These generate*

*extensive lower mantle upwellings at high latitudes until the structures are dispersed by migrating subduction zones after 500 Ma and reassemble at low latitudes.*

*As stated in our reply to Scott King, the differences in the modelled history of basal mantle structures, i.e. their location, size and heterogeneity has implications for modelling the Earth's magnetic field through time. LLSVPs increase the insulation of the core-mantle boundary, decrease the temperature gradient and suppress core-mantle boundary heat flow (Li et al., 2018). Glatzmaier et al. (1999) suggested that the polar core-mantle boundary heat flow may be key to driving magnetic reversal frequency. In contrast, Olson et al. (2010) found that the average polarity reversal frequency is sensitive to the total core-mantle boundary heat flow and to the total heat flow at the equator, while reversal frequency may also increase with the amplitude of the boundary heterogeneity. Our basal mantle structure models could be used to evaluate the effect of alternative plate-mantle models on the spatio-temporal patterns of core-mantle boundary heat flow, and magnetic reversal frequency. Such models could also be used to test the validity of alternative reference frames, in terms of how well modelled magnetic reversal frequencies match observed ones.*

*Further future tests of our absolute reference frames in terms of their suitability as mantle reference frames may include dynamic surface topography models, derived from plate-mantle models. Such models could be compared against geologically mapped continental flooding patterns, following approaches designed to separate effects of eustasy and dynamic topography on continental flooding (e.g., Cao et al., 2019; Müller et al., 2018). In terms of testing alternative orthoversion models, if continents move eastwards after Rodinia breakup, as in our optimised mantle reference frame, one would expect the eastern portions of continents to be flooded first as they move towards dynamic topography lows associated with subduction zones to the east of Rodinia.  In contrast, if continents move westwards after Rodinia breakup as suggested by Mitchell et al. (2012) and implemented as a model with evolving plate boundaries by Cao et al. (2021a), one would expect to see the western portions of continents to be inundated first after Rodinia breakup.*

Authors spend a lot of time comparing their model predictions to present-day mantle structure, as imaged through seismic tomography. I will outline my concerns with the specifics of approach that has been used below. However, in the context of this paper, such comparisons are almost irrelevant: I found them to be a distraction from the paper's main message. Previous studies have shown that deep mantle structure is only sensitive to the past 200-250 Myr of plate motion histories – this has been pretty much clear since the work of McNamara & Zhong (Nature, 2005) and perhaps even Bunge et al. (PTRS, 2002). Given that the different reconstructions examined here are all similar at 200 Myr, there really is not much to be gained (at least in my opinion!) by examining present-day structure. The focus of results and discussion should be more on the differences predicted between each scenario as a function of time, as suggested in point 1 above. The authors almost acknowledge this themselves when stating – "it is noteworthy that the unoptimized model PMAG, not representing a mantle reference frame, reaches an equivalent accuracy to the optimised models OPT1 and OPT2. This reflects that the present-day mantle structure is largely the results of the post- 250 Ma subduction history." I'd suggest removing these comparisons or transferring across to the supplement.

*We agree, and, as also stated in the response to reviewer 1, we have now added an alternative method of assessing the success of the optimised model, following the method used in a recently*

*published paper by Flament et al. (2022). This is a statistical method used to assess how consistent the history of LIP volcanism through time is with modelled mobile basal mantle structures, as compared to fixed ones, over the last billion years.*

Tectonic rules: I realise these have been outlined by Tetley et al. (2019), but it would be good to see a little more of a summary in the current paper, to provide valuable background material. In the context of the current paper, it would also be good to comment on whether such assumptions/rules are reasonable, back to 1 Gyr.

*These rules have not only been outlined by Tetley et al. (2019), but their justification and application in different forms has also been discussed at length in Müller et al. (2019), in the context of designing a mantle reference frame for the last 250 Ma. There is also a fairly comprehensive overview of how the technique works on the related public github repository: https://github.com/EarthByte/optAPM. Seeing that the length of our paper has already increased quite a bit in response to addressing the other aspects of the reviews, especially in adding a LIP volcanism-based method for assessing the quality of the optimised reference frame, we prefer to refrain from inserting additional details about Tetley et al.'s (2019) method here.*

**Minor points:**
Abstract – the approach of Bower et al. (2015) amounts to more than a surface boundary condition (thermal structure, dip angle etc... are also imposed). I would therefore recommend changing line 24 to reflect this. Perhaps "use it as a constraint on mantle flow models", or similar?

*Fixed.*

Line 65 – ... a combination of relative plate motion and *constraints provided by* mantle convection models.

*Fixed.*

Line 80-85 it is stated that the assumption of fixed LLSVPs is based on an apparent correlation between the reconstructed eruption sites of LIPs and kimberlites, from the work of Torsvik, Burke and others. However, using powerful statistical approaches, a number of studies (e.g. Austermann et al. GJI, 2014; Davies et al. EPSL, 2015), have shown that this correlation is not robust, whilst a follow up study by Doubrovine et al. (2016) essentially shows the same (i.e. you cannot conclusively state that plumes are forming at edges over LLSVP interiors). I find it surprising that these studies, which support the authors conclusions for mobile deep mantle structure, are not cited or discussed. Alongside the models presented, these studies provide a solid basis for challenging the fixed LLSVP hypothesis of Torsvik and others.

*Fixed. We have now also added a reference to the recently published work by Flament et al. (2022) who showed that the alignment of LIPs and kimberlites is statistically as consistent with the boundaries and interiors of mobile basal mantle structures as with fixed ones.*

Line 91 – probably fair to cite work by Bull et al. (EPSL, 2009) and Davies et al. (EPSL, 2012) here too.

*Fixed.*

I found the jump in logic from Line 101 to Line 102 hard to follow at first. Do "alternative modes of supercontinent formation" really belong in a subsection on LLSVPs? Having read it a few times, I see the link, but perhaps a separate section, or a sentence explicitly connecting these two aspects, would be helpful.

*We agree, and have now separated this paragraph out under a distinct headline.*

Line 232 – model setup – you limit the age of the lithosphere to 80 Myr when constructing the thermal structure of plate, but still use a half-space model. Why is this? Why not use a plate model, where thickness changes are small beyond this age anyway? Are your results sensitive to this age? If so, it's probably worth explicitly acknowledging that this is the case.

*We have now added this explanatory sentence:*

*This corresponds to a fast and simple implementation of the equivalent of a plate model; for the purpose of our application, the difference to using an actual plate model would be negligible.*

Line 236 – just a flag that the CMB temperature used in these models is very much towards the lower end of current estimates. The Di is also higher than I'd have expected. I'd recommend that authors provide a justification for their choices.

*Di was obtained from $Di = \alpha_0 g_0 R_0 / C_{P_0} = 1.56$, with the reference coefficient of thermal expansivity at the surface $\alpha_0$ = 3 $\times 10^{-5}$ K$^{-1}$. This ignores that α decreases with depth. A depth-average value of α could be used in future work.*

Line 247 – 256 – there seems to be spurious use of bold font in places.
*Fixed.*

Viscosity: I find it difficult to convert that beautiful (!) equation describing your viscosity into an understanding of the range of values in the model and their depth and lateral sensitivity. Could you add a plot showing the depth average and range of values? This will help a reader to place your results in the context of other studies with different rheological approximations.

*We have added a Figure (12b) showing temperature and viscosity profiles at 1000 Ma and the present-day for our model OPT1.*

Comparisons between model predictions and imaged structure: as noted in the main points above, I do not feel that these comparisons add anything to this paper and find them a little distracting. Dropping these comparisons would free up space to discuss your exciting results in more depth. As well as this, I have a major concern with how such comparisons are undertaken. Seismic velocity is non-linearly dependent on temperature, composition and phase. Furthermore, tomographic models have limited and uneven resolution. None of these important factors are considered in the comparisons that you present. Several previous studies (e.g., Bull et al. EPSL 2009; Schuberth et al. G3, 2009; Davies et al. EPSL, 2012) demonstrate that they need to be considered when comparing models with tomography.

In response, we would like to point to this paragraph in the paper:

"We did not attempt to identify a model that falls within range of tomographic models for ($\overline{Acc}$). While the comparison in cluster space is a useful indication of the match, we did not convert mantle flow models from temperature to density, and we did not apply a tomographic operator (which takes the distribution of earthquakes and seismic stations into account) to the results of mantle flow models. Both these steps would affect the present-day match between mantle flow and tomographic models (Davies et al., 2015a; Schuberth et al., 2009). We note that only one tomographic operator is available for such calculations (Ritsema et al., 2011), and that using this operator on the predicted temperature field has a small effect on ($\overline{Acc}$) (Flament et al., 2017). OPT2 is the preferred model overall in this context, as it fits the location of volcanic eruptions and the present-day structure of the lower mantle better than other models."

Just a comment. I REALLY liked Figures 2 and 3. They were very useful for a geodynamicist that is not an expert in plate motion reconstructions. It was valuable to be able to directly compare the different reference frames.

*Thanks!*

Lines 394 – 410. This is a very interesting insight. I don't know the answer of the top of my head and haven't had time to appraise the literature, but are these trends supported by models that examine the evolution of trench retreat under various scenarios (I'm thinking of work by Goes, Garel, Van Hunen, Capitanio, Moresi, Holt, for example)?

*The published models referred to here (we are familiar with most of them) are focused on relatively simple end-member scenarios, usually in Cartesian coordinates, and usually covering relatively short time periods, and we have a hard time seeing any clear connections with these models. We agree that this is likely worth pursuing in more depth, perhaps as a separate paper.*

Line 407 – who doesn't get excited by the "zippy tricentenary"? It's not a term I'll forget in a hurry!

*Indeed!*

Figure 4 – remove duplicate scale bars (unless I'm missing something)?

*Done.*

Line 443 – you mention that there are some periods of relatively large RMS speeds and attribute these to potential artefacts in the reconstruction. Could you say a little more here to help a non-expert? What type of artefacts are these? And why are you confident that they are not present at other times?

*We have added this explanation now:*

*Such artefacts mostly occur as a consequence of the way synthetic, now-subducted ocean crust is reconstructed. These reconstructions assimilate preserved geological evidence related to the types*

*of regional plate boundaries and the timing of the opening or closing of ocean basins, but are nevertheless subject to interpretation, and seafloor spreading rates are often not very well-constrained. As a consequence RMS speed artefacts can arise, which can be addressed in future, improved plate and plate boundary reconstructions.*

Line 470 – you use the term ridges and nodes. What is meant by nodes here? Are they simply ridge intersections? If so, perhaps use ridge intersections instead, or define nodes on first use (given that you also use nodes in a different context elsewhere in manuscript).

*We have deleted the reference to nodes here, which were supposed to refer to ridge intersections, but it is implied that a network of ridges has such intersections, so the term is superfluous.*

Line 490: … basal mantle structure *with upwellings* centered on the north and south pole…

*Fixed.*

Fig. 11 – I could not easily make out the bright red dots. Perhaps enlarge? Or add crosses or similar?

*We have removed the reference to bright red dots. Plumes are simply a part of the temperature structure that is shown, and the reference to dots was confusing, as they are not actually plotted as separate dot symbols.*

Line 643 – it is explicitly mentioned here, but it is also mentioned elsewhere in the paper: short subduction zones have the capacity to roll back faster than long subduction zones. In general, this is true, but I think the reality is a little more nuanced. If the downgoing plates are young, trench retreat is limited, even for short subduction zones. In other words, the magnitude of trench retreat does not only depend on the length of a subduction zone, but also its age (as well as complications arising from overriding plates etc…). This is explicitly covered in a pre-print here https://www.essoar.org/doi/10.1002/essoar.10508606.1. Potentially something that's worth looking at further down the line in your extended reconstructions is whether you see evidence for these dependencies in your reconstructions.

*Yes, we are aware of this behaviour and agree that this deserves to be looked into more deeply further down the line.*

Line 696 – strcturee – structure.

*Fixed*

Line 710-712 – it's probably fair to cite work by Davies et al. (EPSL, 2012) and Bower et al. (G3, 2013) here.

*Done.*

Lines 714-720 – with the comparisons of slab depths it's important to acknowledge that your models do not include phase transitions, which are important in dictating the form of slab transition-zone interaction.

*We have mentioned this now.*

Line 789 – I think a little too much credit is given to the study of Davaille and Romanowicz (2020) here. I'm not denying it's a wonderful study, but it builds on concepts and inferences from many previous studies that are not cited. I would recommend perhaps giving some credit to some earlier work in this area, alongside the work of Davaille and Romanowicz.

*We have cited other papers now as well.*

Lastly, we have also made a change to the author list, which all authors have agreed with, to promote Nicolas Flament to second author, to acknowledge the enormous additional work he has carried out in revising the paper, particularly in carrying the optimised reference frame test using LIPS and kimberlite data.

Flament, N.: Present-day dynamic topography and lower-mantle structure from palaeogeographically constrained mantle flow models, Geophysical Journal International, 216, 2158-2182, 2019.

Flament, N., Bodur, Ö. F., Williams, S. E., and Merdith, A. S.: Assembly of the basal mantle structure beneath Africa, Nature, 603, 846-851, 2022.

Glatzmaier, G. A., Coe, R. S., Hongre, L., and Roberts, P. H.: The role of the Earth's mantle in controlling the frequency of geomagnetic reversals, Nature, 401, 885-890, 1999.

Li, M., Zhong, S., and Olson, P.: Linking lowermost mantle structure, core-mantle boundary heat flux and mantle plume formation, Physics of the Earth and Planetary Interiors, 277, 10-29, 2018.

Müller, R. D., Zahirovic, S., Williams, S. E., Cannon, J., Seton, M., Bower, D. J., Tetley, M. G., Heine, C., Le Breton, E., and Liu, S.: A global plate model including lithospheric deformation along major rifts and orogens since the Triassic, Tectonics, 38, 1884-1907, 2019.

Olson, P. L., Coe, R. S., Driscoll, P. E., Glatzmaier, G. A., and Roberts, P. H.: Geodynamo reversal frequency and heterogeneous core–mantle boundary heat flow, Physics of the Earth and Planetary Interiors, 180, 66-79, 2010.